# A Regret Minimization Framework on Preference Learning in Large Language Models

**Suhwan Kim** [1] [*]  **Taehyun Cho** [1] [*] [†]  **Geon-Hyeong Kim** [2]  **Yu Jin Kim** [2]
**Youngsoo Jang** [3] [‡]  **Moontae Lee** [2] [‡]  **Jungwoo Lee** [1] [4] [‡]

## Abstract

Reinforcement learning with verifiable rewards (RLVR) has enabled progress on reasoning-intensive tasks by relying on task-specific verifiers that provide automated correctness signals. However, many realistic language tasks are difficult to equip with reliable verifiers, motivating a growing reliance on reinforcement learning from human feedback (RLHF). In this setting, we argue that a closer examination of how human feedback should be interpreted is essential. We introduce Regret-based Preference Optimization (**RePO**), which reframes RLHF through *regret minimization* rather than reward maximization. Human preferences are often shaped by *prospective* anticipation of outcomes and *counterfactual* comparisons to alternative behaviors, rather than by immediate, outcome-independent utility. **RePO** captures this structure by modeling preferences as behavior-conditioned assessments of relative suboptimality. Experiments on mathematical reasoning benchmarks and human preference datasets demonstrate consistent performance gains, indicating that **RePO** is an effective and human-aligned approach for training large language models.

## 1. Introduction

Recent advances in large language models (LLMs) have demonstrated substantial improvements on reasoning-intensive tasks, driven in part by reinforcement learning

---

[*]Equal contribution  [†]Work done during internship at LG AI Research. [‡]Equal corresponding authorship. [1]Seoul National University, Seoul, South Korea [2]LG AI Research, Seoul, South Korea [3]UNIST, Ulsan, South Korea [4]HodooAI Labs, Seoul, South Korea. Correspondence to: Youngsoo Jang <youngsoo.jang@unist.ac.kr>, Moontae Lee <moontae.lee@lgresearch.ai>, Jungwoo Lee <junglee@snu.ac.kr>.

*Proceedings of the 43rd International Conference on Machine Learning*, Seoul, South Korea. PMLR 306, 2026. Copyright 2026 by the author(s).

techniques that leverage explicit supervision signals. In particular, RLVR has enabled notable progress in domains such as mathematical reasoning and code generation, where task-specific verifiers can reliably determine correctness. By providing automated and scalable supervision, such pipelines have led to significant gains in accuracy and consistency.

Despite its success, RLVR addresses only a limited subset of realistic language tasks. In many practically important settings—such as safety-critical decision making, customer-facing service interactions, and educational reasoning tasks—reliable verifiers are difficult or impossible to specify. As a result, learning systems are increasingly forced to rely on reinforcement learning from human feedback (RLHF), where supervision is provided in the form of preferences, comparisons, or qualitative judgments, without explicit reward signals supplied by verifiers. In this setting, we do not address the scarcity of supervision itself, but instead focus on *how human feedback should be interpreted*.

A distinctive feature of human judgment is inherently *prospective* and *counterfactual*, rather than local or absolute. When evaluating partial or intermediate reasoning steps, humans do not assess them in isolation, but form judgments by anticipating how the reasoning is likely to unfold and by comparing the observed behavior against plausible alternatives that could have been taken. Consequently, preferences over intermediate trajectories already reflect expectations about future outcomes and imagined counterfactual continuations. This characterization is supported by extensive findings in cognitive psychology, which show that humans naturally engage in mental simulation of future outcomes (Tversky et al., 1982) and evaluate decisions through counterfactual comparisons with unrealized alternatives (Kahneman & Miller, 1986; Byrne, 2016).

Nevertheless, most existing preference-based learning methods interpret such feedback through the lens of reward maximization, treating preferences as proxies for immediate or accumulated utility assigned to the observed segment. While convenient, this interpretation overlooks the evaluative structure implicit in human judgment. In particular, it ignores the fact that preferences often reflect relative assessments of suboptimality—how a behavior compares to

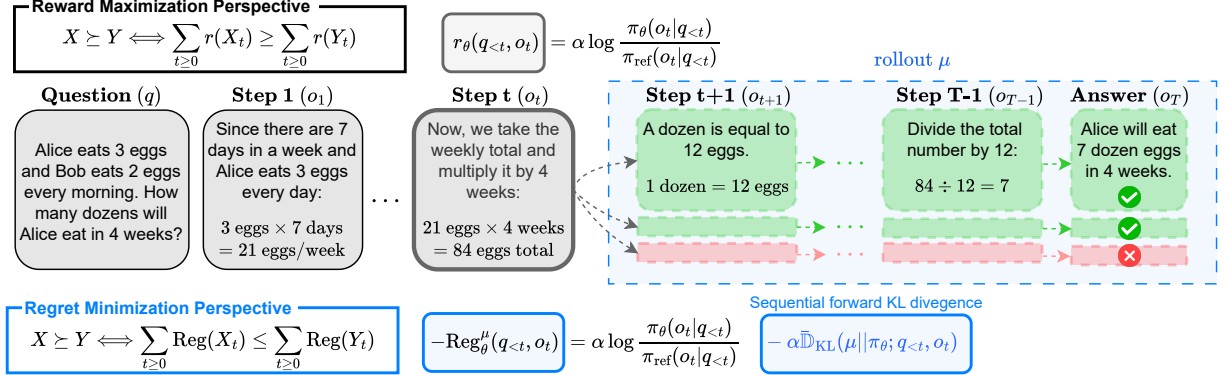

*Figure 1.* Comparison between reward-maximization and regret-minimization reasoning. In reward maximization, evaluation is local to each step and depends only on immediate rewards. In contrast, regret minimization performs prospective reasoning, continuing the trajectory forward to a verifiable future state, and then applies a retrospective reassessment of earlier steps using the realized outcome.

what could have been done instead—rather than absolute notions of goodness. This issue also arises in recent reasoning pipelines that provide supervision at intermediate steps, for example through rollout-based soft labels (Wang et al., 2024). While such signals are often treated as local rewards, they implicitly depend on how the reasoning is expected to continue, and thus cannot be understood as outcome-independent utility signals.

Motivated by these considerations, our work makes the following contributions. First, we revisit preference-based learning in RLHF from a regret-minimization perspective, arguing that human feedback is more naturally understood as relative judgments conditioned on anticipated outcomes and implicit alternatives, rather than as signals for reward maximization. Second, we formalize this perspective within a KL-regularized reinforcement learning framework and introduce Regret-based Preference Optimization (**RePO**), which admits a closed-form policy update compatible with direct preference optimization. Finally, we demonstrate through experiments on mathematical reasoning and human preference benchmarks that **RePO** leads to effective and more human-aligned training of LLMs.

## 2. Related Works

**Reinforcement Learning from Human Feedback.** Reinforcement Learning from Human Feedback (RLHF) is the dominant paradigm for aligning large language models with human preferences (Christiano et al., 2017; Stiennon et al., 2020). The standard RLHF pipeline consists of supervised fine-tuning (SFT), reward modeling, and policy optimization, with Proximal Policy Optimization (PPO) (Schulman et al., 2017) being the most widely used optimizer due to its stability in high-dimensional action spaces and its trust-region-like objective clipping. Nevertheless, the reliance on explicit reward models and the computational burden of

maintaining multiple models have motivated the development of more direct preference-based optimization methods.

**Direct Preference Optimization and Variants.** Direct Preference Optimization (DPO) has motivated a range of variants that refine its preference-based objective for language modeling. Token-level DPO (**TDPO**) (Zeng et al., 2024) extends DPO to token-level, multi-step trajectories and introduces forward-KL constraints at each token to improve alignment while preserving generation diversity. Regularized Preference Optimization (**RPO**) (Liu et al., 2024) mitigates over-optimization by regularizing preference optimization with an SFT-based term, offering theoretical guarantees for improved stability. Identity Preference Optimization (**IPO**) (Azar et al., 2024) avoids the Bradley-Terry assumption of DPO by directly optimizing preference likelihoods, maintaining effective KL regularization to prevent over-optimization from reward mis-specification.

**Beyond Reward Maximization.** Recent work increasingly recognizes that human feedback is inherently comparative rather than absolute, motivating alternatives to scalar reward maximization. Contrastive Preference Learning (**CPL**) (Hejna et al., 2023) avoids explicit reward modeling by defining preferences through optimal advantage, focusing on relative utility between trajectories. Policy-labeled Preference Learning (**PPL**) (Cho et al., 2025) highlights that using optimal advantage may implicitly conflate environmental uncertainty with policy behavior, which can lead to likelihood mismatch under heterogeneous or off-policy data. By conditioning preferences on the behavior policy, **PPL** introduces a regret-based formulation that enables stable learning from heterogeneous preference datasets. Kahneman–Tversky Optimization (**KTO**) (Ethayarajh et al., 2024) takes a complementary perspective by incorporating prospect-theoretic preference responses, achieving robust alignment directly

from binary labels. We provide a detailed comparison and discussion of the conceptual and technical differences between our work and prior approaches in Appendix B.

## 3. Preliminaries

We consider a reward-free formulation of a Markov decision process, where learning is driven entirely by preference feedback rather than explicitly defined rewards. Rather than treating rewards as a primitive signal, we focus on how human feedback is interpreted and incorporated into policy learning.

**KL-regularized RL.** Formally, we work within a *step-level* KL-regularized RL framework, which regularizes updates to remain close to a reference policy while allowing preferences to guide learning, i.e.,

$$\pi^*_{\text{KL-reg}} := \arg\max_{\pi_\theta} \mathbb{E}_{q\sim\mathcal{D}, o_t\sim\pi_\theta(\cdot|q_{<t})}\Big[\sum_{t\leq T}\gamma^t\Big(r(q_{<t}, o_t)$$
$$- \alpha\mathbb{D}_{\text{KL}}\big(\pi_\theta(\cdot|q_{<t})||\pi_{\text{ref}}(\cdot|q_{<t})\big)\Big)\Big] \quad (1)$$

where $\mathbb{D}_{\text{KL}}(\mu(\cdot)||\nu(\cdot)) = \mathbb{E}_\mu[\log(\mu(\cdot)/\nu(\cdot))]$.

Here, we define a chain-of-thought (CoT) output as a *full trajectory* $q_{\leq T} := (q, o_1, \ldots, o_T) \in \mathcal{S}$, where $q$ denotes the input query and $(o_1, \ldots, o_T)$ are intermediate reasoning steps. The final element $o_T$ corresponds to the terminal output. For any $t \leq T$, we define the corresponding *context* $q_{\leq t} := (q, o_1, \ldots, o_t)$, which represents a partial chain-of-thought. We consider a stochastic policy $\pi_\theta$, parameterized by $\theta$, which induces a distribution over full trajectories $q_{\leq T}$ as well as over their contexts $q_{\leq t}$. When a task-specific verifier is available, we let $o_T^\star \in \mathcal{O}^\star$ denote a *verifier-accepted* terminal output.

Human annotators or AI evaluators are provided with complete trajectories and asked to express preferences over contexts. Concretely, we observe pairwise comparisons of the form $(q_{\leq t}^+, q_{\leq t}^-)$, indicating that context $q_{\leq t}^+$ is preferred over $q_{\leq t}^-$, denoted by $q_{\leq t}^+ \succ q_{\leq t}^-$.

**Score-based Preference Model.** To implement the preference model using a neural network, we parametrize the score function as $S_\theta$, and the model is trained by minimizing the cross-entropy loss between its predictions and the preference labels in the dataset $\mathcal{D}$, as follows:

$$\mathbb{P}_{S_\theta}[q_{\leq t}^+ \succ q_{\leq t}^-] = \sigma\Big(S_\theta(q_{\leq t}^+) - S_\theta(q_{\leq t}^-)\Big),$$
$$\mathcal{L}(S_\theta; \mathcal{D}) = -\mathbb{E}_{(q^+, q^-)\sim\mathcal{D}}\Big[\log \mathbb{P}_{S_\theta}[q_{\leq t}^+ \succ q_{\leq t}^-]\Big] \quad (2)$$

where $\sigma(x) = 1/(1 + e^{-x})$. For notational simplicity, we write the dataset expectation as $\mathbb{E}_{(q^+, q^-)\sim\mathcal{D}}$ as $\mathbb{E}_\mathcal{D}$.

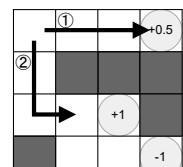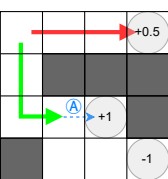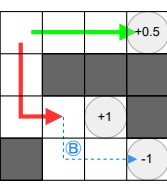

*Figure 2.* Humans evaluate partial trajectories prospectively: even when a trajectory is incomplete, human mentally extend it toward a plausible future and judge the current segment in light of that anticipated outcome. In the left illustration, segment ② is on an optimal path while segment ① leads to a suboptimal outcome. If the evaluator anticipates a favorable continuation Ⓐ, segment ② is preferred; if a pessimistic continuation is imagined Ⓑ, segment ① may instead be preferred. However, current RLHF interprets such feedback myopically at the segment level, resulting in a mechanism-level mismatch with human judgment.

A common instantiation of the score function is the immediate-reward formulation, in which higher cumulative reward over a given context implies a higher probability of being preferred. Rafailov et al. (2024) show, in a contextual-bandit setting, that the reward can be expressed as the relative log-likelihood of a policy with respect to a reference policy, leading to a closed-form policy update known as Direct Preference Optimization (DPO). Around the same time, An et al. (2023) define the score in terms of action-space distances for preference-based policy optimization in robotic manipulation tasks.

## 4. A Regret Minimization Framework on Preference Learning

### 4.1. Limitations of Reward Maximization Framework

In reasoning-centric LLMs, preference feedback is commonly incorporated through the Bradley–Terry model, effectively interpreting preferences as rewards—or partial sums thereof—for supervised learning. To reduce the cost of human annotation, many recent studies replace direct labeling with automatic annotators. In these pipelines, labels for intermediate steps are often inferred from the accuracy obtained by rolling out from that step, a practice first popularized by Math-Shepherd (Wang et al., 2024). Despite its intuitive appeal, this mechanism reveals structural inconsistencies upon closer inspection. Importantly, these issues are not specific to any particular annotation pipeline or heuristic, but arise from the way preferences are fundamentally interpreted as immediate rewards.

**Misalignment with Prospective judgment.** In standard RL, rewards are defined as immediate utilities assigned to state–action pairs, independent of how future outcomes ultimately unfold. As a result, incomplete trajectory segments that have not yet reached a terminal state are evaluated solely based on the rewards obtained so far.

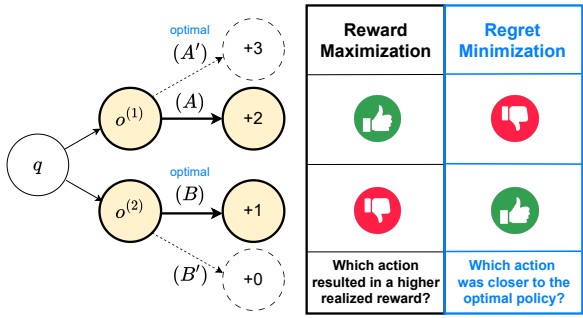

*Figure 3.* Reward maximization evaluates actions by realized outcomes, whereas regret minimization evaluates their proximity to optimal behavior under counterfactual alternatives.

In Figure 2, segment ② receives zero reward throughout and is therefore indistinguishable from, or even inferior to, segment ① under a reward maximization criterion. However, human evaluators often prefer segment ② when they anticipate a plausible favorable continuation, as illustrated by outcome Ⓐ. This preference arises despite the absence of immediate reward differences, indicating that humans assess partial trajectories prospectively by mentally extending them toward likely future outcomes. When a pessimistic continuation such as Ⓑ is anticipated, the preference may instead reverse. This dependence on imagined futures reveals a fundamental limitation of reward-based RLHF: rewards defined as immediate utilities fail to capture the prospective judgment mechanism through which humans assign preferences to intermediate behavior.

**Misalignment with Counterfactual Thinking.** Human decision-making is inherently *counterfactual*: evaluations are often formed by comparing the realized outcome with plausible alternatives that could have occurred under different choices (Byrne, 2016). Figure 3 illustrates how this mode of evaluation fundamentally departs from reward maximization. Under a reward-based criterion, decision $(A)$ appears preferable to decision $(B)$ because it yields a higher realized payoff. However, human evaluators do not assess decisions solely based on realized outcomes. When counterfactual alternatives $(A')$ and $(B')$ are taken into account, preferences can reverse: if decision $(B)$ would have led to consistently favorable outcomes across plausible alternatives, while decision $(A)$ admits unfavorable counterfactual realizations, evaluators often prefer $(B)$ despite its lower realized reward.

Crucially, this observation suggests that human preferences are not confined to realized trajectories, but are instead shaped by comparisons against counterfactual best outcomes, and are therefore inherently tied to relative suboptimality. Returning to the setting where no explicit reward is defined, a natural question arises: *how should such feedback be interpreted and modeled?* In the next section, we introduce *regret minimization framework* that captures both the prospective and counterfactual nature of human feedback.

## 4.2. Interpreting Human Preferences through the Lens of Regret

As human feedback is inherently prospective and counterfactual, it is more naturally interpreted through regret rather than reward. Unlike reward, which assigns instantaneous utility to realized actions, regret measures suboptimality relative to what could have been achieved under an optimal decision at the same context. From this perspective, preference feedback guides decisions toward reducing regret with respect to implicitly imagined alternatives.

Regret has been extensively studied in online reinforcement learning as a principled notion for characterizing decision quality and learning efficiency (Lattimore & Szepesvári, 2020; Szepesvári, 2022). Importantly, its definition aligns closely with how humans evaluate decisions: by comparing realized behavior against counterfactual best outcomes under the same conditions. This interpretation makes regret a natural abstraction for modeling human preferences when explicit reward signals are absent.

Motivated by this interpretation, we adopt negative *regret* as the score function for preference learning in LLMs.[1] Intuitively, if preferences encode relative suboptimality, then minimizing regret provides a direct operational objective. Concretely, by instantiating the preference score in Equation (2) with negative regret, we introduce **Re**gret-based **P**reference **O**ptimization (**RePO**), which admits a closed-form policy update analogous to DPO while explicitly accounting for behavior-dependent deviation.

$$\text{Reg}^{\mu}_{\pi^*}(q_{<t}, o_t) := V^{\pi^*}(q_{<t}) - Q^{\mu}(q_{<t}, o_t)$$
$$\stackrel{\text{(Thm 5.4)}}{=} -\alpha\Big(\log \frac{\pi^*(o_t|q_{<t})}{\pi_{\text{ref}}(o_t|q_{<t})} - \bar{\mathbb{D}}_{\text{KL}}(\mu||\pi^*; q_{<t}, o_t)\Big)$$

where $\mu$ denotes the behavior policy that generates the trajectory leading to the context $q_{<t}$, and $\bar{\mathbb{D}}_{\text{KL}}$ is the sequential KL divergence defined in Theorem 5.4. In our setting, the behavior policy denotes the policy actually executed during rollout toward a plausibly verifiable state, and it may differ from the optimal policy. The RePO objective is detailed in Appendix D.3.

## 5. Theoretical Results

This section presents the theoretical results required to perform DPO within a discounted KL-regularized RL framework. We begin by characterizing the relationship between the optimal policy and the reward function. To this end,

---

[1]The term *regret* here differs slightly from its standard usage in online RL, where it denotes the cumulative suboptimality gap accumulated over *episodes*. In our offline preference learning setting, we instead define regret at the *trajectory-level* as the sum of per-step regrets. For brevity, we use regret to refer to this step-wise quantity throughout the paper.

we define equivalence classes of rewards and $Q$-functions, and establish structural conditions under which a given policy becomes optimal. Building on this characterization, we derive a closed-form expression of the $Q$-function associated with a behavior policy $\mu$. Finally, we formulate a DPO objective grounded in the regret minimization framework.

**Lemma 5.1.** *Let the Q-function of policy $\mu$ be defined as*

$$Q^\mu(q_{<t}, o) := r(q_{<t}, o) +$$
$$\mathbb{E}_{\mathbb{P}^\mu}\Big[\sum_{l>0} \gamma^l\big(r_{t+l} - \alpha\mathbb{D}_{KL}(\mu(\cdot|q_{<t+l})||\pi_{ref}(\cdot|q_{<t+l})))\big)\Big].$$

*Define value function $V^\mu$ satisfying the Bellman equation*

$$V^\mu(q_{<t}) = \mathbb{E}_\mu[Q^\mu(q_{<t}, o) - \alpha\mathbb{D}_{KL}(\mu(\cdot|q_{<t})||\pi_{ref}(\cdot|q_{<t}))]$$
$$Q^\mu(q_{<t}, o) = r(q_{<t}, o) + \gamma\mathbb{E}_{\mathbb{P}^\mu}[V^\mu(q_{<t+1})].$$

*Then the optimal value function and the optimal policy satisfies the following relation:*

$$V^{\pi^*}(q_{<t}) = \alpha\log\mathbb{E}_{o\sim\pi_{ref}}\Big[\exp\Big(\frac{1}{\alpha}Q^{\pi^*}(q_{<t}, o)\Big)\Big],$$
$$\frac{\pi^*(o|q_{<t})}{\pi_{ref}(o|q_{<t})} = \exp\Big(\frac{1}{\alpha}(Q^{\pi^*}(q_{<t}, o) - V^{\pi^*}(q_{<t}))\Big).$$

Lemma 5.1 characterizes the value function associated with $Q$-functions satisfying Objective (1) as a $\pi_{\text{ref}}$-weighted energy-based model, which enables a Bellman update. This characterization naturally generalizes the maximum entropy framework of Haarnoja et al. (2017) to the KL-regularized reinforcement learning formulation considered in this work.

**Definition 5.2** (KL-regularized version of Cho et al. (2025)). For a given reference policy $\pi_{\text{ref}}$, the set of reward functions where $\pi^*$ is $\alpha$-optimal is defined as $(\alpha, \pi^*)$-*equivalence class of reward function*, denoted by $\mathcal{R}_{\alpha,\pi^*}$. For every policy $\pi$, the set of $Q^\pi$-function generated by any reward function $r_{\alpha,\pi^*} \in \mathcal{R}_{\alpha,\pi^*}$ is defined as the $(\alpha, \pi^*)$-*equivalence class of $Q^\pi$-function*, denoted by $\mathcal{Q}^\pi_{\alpha,\pi^*}$.

Definition 5.2 indicates that a reward function class $\mathcal{R}$ or a $Q^\pi$-function class $\mathcal{Q}^\pi$ can be partitioned based on the $\alpha$-optimal policy $\pi^*$. For notational simplicity, we denote the ground truth reward function corresponding to the $\alpha$-optimal policy $\pi^*$ as $r_*$ and the $Q^\pi$-function induced by $r_*$ as $Q^\pi_*$, simplifying the subscript to $*$.

**Lemma 5.3** (Structural Condition for $\alpha$-optimality). *An optimal Q-function where $\pi^*$ is $\alpha$-optimal have a one-to-one correspondence with a state-dependent function $\beta : \mathcal{S} \to \mathbb{R}$, defined as follows:*

$$\mathcal{R}^{\pi^*} = \Big\{ r_*(q_{<t}, o) = \alpha\log\frac{\pi^*(o|q_{<t})}{\pi_{ref}(o|q_{<t})} + \beta(q_{<t})$$
$$- \gamma\mathbb{E}_{\mathbb{P}}[\beta(q_{<t+1})]\Big\},$$
$$\mathcal{Q}^{\pi^*} = \Big\{ Q^{\pi^*}_*(q_{<t}, o) = \alpha\log\frac{\pi^*(o|q_{<t})}{\pi_{ref}(o|q_{<t})} + \beta(q_{<t})\Big\}.$$

Lemma 5.3 shows that the $(\alpha, \pi^*)$-equivalence class of optimal $Q$-functions admits a unique decomposition into an action-dependent log-probability term, $\alpha\log\frac{\pi^*(o|q_{<t})}{\pi_{\text{ref}}(o|q_{<t})}$, and a state-dependent shaping function $\beta(q_{<t})$. While the log-probability term has been widely studied—most notably in Rafailov et al. (2024)—the role of $\beta(\cdot)$ has largely been overlooked. In contextual bandit settings, $\beta(\cdot)$ can be absorbed into a constant due to the absence of state transitions. However, in token- or step-level MDPs, this cancellation no longer holds: the shaping term propagates through future states, inducing state-dependent shifts in the $Q$-function and affecting the evaluation of intermediate decisions. From a reward-maximization perspective, this implies an implicit dependence on $\beta(\cdot)$, which must be specified or estimated.

**Effect of $\beta(\cdot)$ under finite data.** Lemma 5.3 implies that reward-maximization objectives are not identifiable up to additive shaping. In particular, constant choices of $\beta(\cdot)$ correspond to reward translations that preserve $\alpha$-optimal policies. Although such invariance is benign asymptotically, it can noticeably affect learning under finite data: large $\beta(\cdot)$ biases learning toward conservative, in-distribution behavior, whereas small $\beta(\cdot)$ reduces penalties for distributional shift and may encourage OOD actions. Thus, while policy-invariant in theory, reward translations can influence exploration-exploitation tradeoff in practice.

A common response is to impose anchoring assumptions, such as reward normalization (Guo et al., 2025) or a zero-reward "I don't know" action (Kalai et al., 2025) (see Appendix C). These approaches, however, rely on externally chosen criteria to fix $\beta(\cdot)$. In contrast, the regret-minimization framework developed in this work is invariant to reward translations by construction, enabling analysis without committing to a specific reward scale. Determining principled ways to select $\beta(\cdot)$ within reward-maximization frameworks remains an important open problem. Further discussion is provided in Appendix C.

**Theorem 5.4** (KL-divergence Policy Deviation Theorem). *If a policy $\pi^*$ is $\alpha$-optimal, then for any policy $\mu$,*

$$Q^{\pi^*}_*(q_{<t}, o) - Q^\mu_*(q_{<t}, o) = \alpha\bar{\mathbb{D}}_{KL}(\mu \,\|\, \pi^*; q_{<t}, o),$$

*where the **sequential forward KL-divergence** is defined as*

$$\bar{\mathbb{D}}_{KL}(\mu \,\|\, \pi^*; q_{<t}, o)$$
$$:= \mathbb{E}_{\tau\sim\mathbb{P}^\mu}\Big[\sum_{l>0}\gamma^l\mathbb{D}_{KL}\big(\mu(\cdot|q_{<t+l}) \,\|\, \pi^*(\cdot|q_{<t+l})\big)\Big].$$

*Here, $\mathbb{P}^\mu$ denotes the distribution of trajectories induced by policy $\mu$ and transition kernel $\mathbb{P}$, starting from the initial context-action pair $(q_{<t}, o)$.*

Theorem 5.4 establishes that, for any behavior policy $\mu$, the difference between its $Q$-function and the optimal $Q$-function can be expressed exactly as a discounted sequential

forward KL divergence from $\mu$ to the optimal policy $\pi^*$. In particular, this difference depends only on the future rollout induced by $\mu$ starting from the given context and output, and admits a trajectory-level interpretation through $\bar{\mathbb{D}}_{\text{KL}}(\mu \| \pi^*; q_{<t}, o)$. Intuitively, the sequential KL term measures the cumulative deviation of $\mu$ from $\pi^*$ along future states visited under $\mu$, discounted over time.

**Lemma 5.5** (Regret Invariance under $(\alpha, \pi^*)$-Equivalent Rewards). *For any reward $r_* \in \mathcal{R}^{\pi^*}$ and any behavior policy $\mu$, the regret incurred at context $q_{<t}$ by selecting output $o$ admits the following representation:*

$$Reg_{\pi^*}^{\mu}(q_{<t}, o) := V^{\pi^*}(q_{<t}) - Q^{\mu}(q_{<t}, o)$$
$$= -\alpha \left( \log \frac{\pi^*(o \mid q_{<t})}{\pi_{\text{ref}}(o \mid q_{<t})} - \bar{\mathbb{D}}_{KL}(\mu \| \pi^*; q_{<t}, o) \right),$$

*where the expression is invariant to the choice of the state-dependent shaping function $\beta(\cdot)$.*

This decomposition reveals that regret consists of two complementary components: (i) a local preference term that favors outputs with higher relative likelihood under the optimal policy, which coincides with the DPO objective, and (ii) a long-horizon term that penalizes the cumulative discrepancy between $\mu$ and the $\pi^*$ along future rollouts. As a result, minimizing regret encourages learning that simultaneously increases the relative likelihood of preferred outputs and reduces long-term policy deviation from optimal behavior.

Under this view, DPO implicitly treats the behavior policy as on-policy, assuming that preferences are generated from rollouts of the learned policy itself. This assumption breaks down in off-policy or offline settings, where preference data are collected under heterogeneous behavior policies. As shown in Cho et al. (2025), this *likelihood mismatch* leads to *MDP indistinguishability*, under which different MDPs become indistinguishable from preference data alone, resulting in suboptimal learning. The regret-based formulation explicitly accounts for this mismatch by incorporating behavior policy information and cumulative future deviation.

## 6. Practical Implementation

**Deterministic Pseudo-labels.** Ideally, the behavior policy can be labeled jointly during the data generation process, making it fully accessible. However, in practice, this assumption may not hold when working with pre-collected offline datasets or when loading the behavior policy model is computationally prohibitive. As a practical alternative, we introduce a *deterministic pseudo-labeling* that alleviates this computational burden and yields a feasible algorithm, which we refer to as `RePO_det`. `RePO_det` replaces the behavior policy $\mu$ with a Dirac distribution concentrated at the observed output $o_t$, effectively treating each sample as if it were generated by a deterministic policy.

**Sample-based Estimation of Regret.** In practice, computing the sequential KL divergence exactly requires rolling out each behavior policy from every intermediate pair $(q_{<t}, o_t)$ until termination, which incurs substantial computational and memory overhead. To obtain a tractable estimator, we replace the discounted infinite-horizon sum with an undiscounted finite-horizon approximation of length $T - t$, normalize the scale across timesteps, and reuse the observed segments $q^+$ and $q^-$ as single rollouts of $\mu^+$ and $\mu^-$, respectively. This yields the following empirical regret estimator:

$$S^{\text{RePO}} := \widehat{Reg}_{\pi_\theta}^{\mu}(q_{<t}, o_t) = -\alpha \Big( \log \frac{\pi_\theta(o_t|q_{<t})}{\pi_{\text{ref}}(o_t|q_{<t})}$$
$$+ \frac{1}{T-t} \sum_{1 \leq l \leq T-t} \log \frac{\pi_\theta(o_{t+l}|q_{<t+l})}{\mu(o_{t+l}|q_{<t+l})} \Big).$$

This approximation can be interpreted as a Monte Carlo estimator of the sequential KL term truncated to a finite horizon. When the behavior policy is unavailable or too costly to evaluate, we adopt a deterministic pseudo-labeling scheme in which $\mu(\cdot \mid q) = \delta_{o_t}$. In this case, the regret estimator simplifies to

$$S^{\text{RePO\_det}} := \widehat{Reg}_{\pi_\theta}(q_{<t}, o_t) = -\alpha \Big( \log \frac{\pi_\theta(o_t|q_{<t})}{\pi_{\text{ref}}(o_t|q_{<t})}$$
$$+ \frac{1}{T-t} \sum_{1 \leq l \leq T-t} \log \pi_\theta(o_{t+l}|q_{<t+l}) \Big).$$

**Lemma 6.1.** *Assume $\epsilon \geq 0$ where*

$$\epsilon := \mathbb{D}_{KL}(\mu(\cdot|q_{<t})||\pi_{ref}(\cdot|q_{<t})) - \mathbb{D}_{KL}(\mu(\cdot|q_{<t})||\pi^*(\cdot|q_{<t})).$$

*Then, for any verifier-accepted output $o_T^\star$,*

$$\widehat{Reg}_{\pi^*}^{\mu}(q_{<T}, o_T^\star) \leq \mathbb{E}_\mu[\widehat{Reg}_{\pi^*}^{\mu}(q_{<t}, \cdot)].$$

Lemma 6.1 formalizes a simple structural property of human feedback under a mild and interpretable assumption. Specifically, the condition $\epsilon \geq 0$ requires that the behavior policy $\mu$ is closer to the optimal policy $\pi^*$ than to the reference policy $\pi_{\text{ref}}$ in KL divergence, which naturally holds for trajectories that reach a verifier-accepted outcome.

**Inductive Bias of Regret.** Under this condition, Lemma 6.1 shows that regret at the terminal accepted state is upper bounded by the expected regret over intermediate, partially observed contexts. Intuitively, masking parts of a successful trajectory introduces uncertainty about future completion, leading to a pessimistic evaluation despite identical underlying behavior. Consequently, incomplete trajectories are systematically judged as worse than fully revealed ones, revealing an inductive bias inherent in regret-based feedback. In Section 7.2, we empirically demonstrates that this inductive bias translates into improved sample efficiency of proposed method.

*Table 1.* AlpacaEval2 (Dubois et al., 2024), Arena-Hard (Li et al., 2024), and MT-Bench (Zheng et al., 2023) results on Qwen3-1.7B/4B. Best results are in **bold**, with second-best underlined.

| Method | AlpacaEval 2 | | Arena-Hard | MT-Bench | |
|---|---|---|---|---|---|
| | LC (%) | WR (%) | WR (%) | GPT-4.1 | GPT-5.1 |
| Qwen3-1.7B-Base | | | | | |
| Base | 8.27 | 6.05 | 10.4 | 5.09 | 4.41 |
| DPO | 23.90 | 25.84 | 23.4 | 6.00 | 4.81 |
| IPO | 24.46 | 26.37 | 21.9 | 6.56 | 5.10 |
| RPO | 18.63 | 15.47 | 17.7 | 5.98 | 5.13 |
| KTO | 34.73 | 38.93 | **30.4** | **6.92** | 5.32 |
| TDPO | 12.24 | 10.72 | 14.5 | 5.56 | 4.73 |
| RePO | **36.61** | **43.66** | 27.1 | 6.88 | **5.43** |
| RePO_det | 34.95 | 41.42 | 26.6 | 6.89 | 5.16 |
| Qwen3-4B-Base | | | | | |
| Base | 12.80 | 11.62 | 25.4 | 5.56 | 4.83 |
| DPO | 32.89 | 33.92 | 44.5 | 6.79 | 5.74 |
| IPO | 36.43 | 38.63 | 47.8 | 7.43 | 6.14 |
| RPO | 29.51 | 28.23 | 41.9 | 7.05 | 6.13 |
| KTO | 52.31 | 55.78 | **63.9** | **8.22** | 6.93 |
| TDPO | 17.97 | 17.08 | 30.9 | 6.24 | 5.38 |
| RePO | **55.08** | **60.12** | 60.1 | 8.09 | 6.78 |
| RePO_det | 51.66 | 55.53 | 59.9 | 8.18 | **6.97** |

*Table 2.* Benchmark results for math-reasoning tasks using Qwen3-1.7B/4B across different preference optimization methods. Best results are in **bold**, with second-best underlined.

| Method | GSM8k | MATH | MATH500 | AMC23 | Minerva |
|---|---|---|---|---|---|
| Qwen3-1.7B-Base | | | | | |
| Base | 61.71 | 48.50 | 48.60 | 30.00 | 9.60 |
| DPO | 77.33 | 53.44 | 52.80 | 32.50 | 16.91 |
| RPO | 69.07 | 50.32 | 51.40 | 30.00 | 13.24 |
| IPO | 79.45 | 51.76 | 53.40 | 20.00 | 16.54 |
| KTO | 79.68 | 54.42 | 56.60 | **35.00** | 17.28 |
| TDPO | 64.06 | 48.80 | 52.20 | 25.00 | 9.93 |
| RePO | 80.52 | 54.50 | **57.40** | 30.00 | 20.59 |
| RePO_det | **80.74** | **54.84** | 54.40 | 25.00 | **25.74** |
| Qwen3-4B-Base | | | | | |
| Base | 78.77 | 61.20 | 64.20 | 32.50 | 19.90 |
| DPO | 87.87 | 56.66 | 57.80 | 35.00 | 27.21 |
| RPO | 90.30 | 63.44 | 66.80 | 47.50 | 22.79 |
| IPO | 88.86 | 58.36 | 57.40 | 45.00 | **27.57** |
| KTO | 90.83 | **67.38** | **67.60** | **55.00** | 25.74 |
| TDPO | 90.67 | 62.76 | 64.8 | 47.50 | 24.26 |
| RePO | 90.60 | 65.54 | 66.20 | 42.50 | 22.43 |
| RePO_det | **91.05** | 65.72 | 65.40 | 47.50 | 23.50 |

# 7. Experiments

We evaluate **RePO** on both human preference alignment and mathematical reasoning tasks. Our experiments address three questions: (i) whether **RePO** improves performance on both benchmarks, (ii) whether **RePO** still remains effective when behavior-policy information is unavailable, and (iii) whether **RePO** internalizes human inductive biases without explicit data augmentation.

## 7.1. Does RePO Learn Effectively on Human Preference and Mathematical Reasoning Benchmarks?

**Human Preference Alignment Results.** We evaluate **RePO** on human preference alignment tasks using a synthetic dataset constructed following the UltraFeedback protocol (Cui et al., 2024). Experiments are conducted with Qwen3-1.7B-Base and Qwen3-4B-Base models (Yang et al., 2025), fine-tuned using LoRA (Hu et al., 2022). Preference data are generated by four Qwen-series models, with token-level log probabilities of the behavior policy retained, resulting in approximately 16K preference pairs. We report length-controlled win rate (LC) and raw win rate (WR) on AlpacaEval 2 using GPT-4 Turbo as the judge, and additionally evaluate on Arena-Hard and MT-Bench using GPT-4.1 and GPT-5.1. See Appendix D for details.

As shown in Table 1, **RePO** consistently outperforms existing baselines across both model scales. Compared to DPO, **RePO** achieves stronger performance by explicitly accounting for behavior-policy-dependent sequential KL divergence, rather than assuming on-policy data. **RePO** also demonstrates competitive performance relative to **KTO**, despite relying on a fundamentally different modeling assump-

tion. While **KTO** incorporates prospect-theoretic loss aversion in preference responses, **RePO** improves alignment by explicitly interpreting preferences through regret minimization under prospective and counterfactual reasoning. Overall, these results highlight the importance of modeling how human feedback is generated, particularly the role of behavior policies and future rollouts.

**Mathematical Reasoning Results.** We evaluate **RePO** on mathematical reasoning tasks using preference data constructed from GSM8K (Cobbe et al., 2021) and MATH (Hendrycks et al., 2021). For each query, we generate eight candidate responses using Qwen2.5-7B-Math-Instruct (Yang et al., 2024). Preference pairs are constructed only for queries that admit both correct and incorrect solutions, where correct solutions are treated as preferred and incorrect ones as non-preferred. Log probabilities are recorded for each generated response, yielding approximately 69K preference pairs for training. Evaluation is conducted on both in-domain benchmarks (GSM8K, MATH, and MATH500) and out-of-domain benchmarks (AMC23 and MinervaMath (Lewkowycz et al., 2022)), allowing us to assess both generalization within the training distribution and robustness under distributional shift. See Appendix D for details.

As reported in Table 2, **RePO** consistently outperforms DPO and its variants across backbone scales and evaluation settings, even when supervision is derived from verifier-based correctness rather than human preferences. This result suggests that regret minimization provides effective learning signals beyond subjective human feedback, and that interpreting preferences through proximity to an optimal policy serves as a useful inductive bias for mathematical reasoning.

*Table 3.* Performance on offline datasets with unknown behavior policies. Best results are in **bold**, with second-best underlined.

| Method | GSM8k | MATH | MATH500 | AMC23 | Minerva |
|--------|-------|------|---------|-------|---------|
| | | | Llama3.1-8B-Base | | |
| Base | 20.02 | 10.14 | 9.20 | 2.50 | 8.46 |
| DPO | 44.50 | 13.46 | 11.80 | 10.00 | 9.19 |
| RPO | 28.43 | 12.40 | 12.60 | 7.50 | 8.09 |
| IPO | 45.79 | 13.44 | 12.00 | 5.00 | 8.09 |
| TDPO | 22.44 | 11.66 | 8.40 | 5.00 | 6.62 |
| KTO | 61.56 | 19.50 | 19.80 | 12.5 | **11.76** |
| RePO_det | **62.17** | **21.30** | **21.60** | 10.00 | 9.19 |
| | | | Llama3.1-8B-Instruct | | |
| Base | 81.96 | 42.22 | 38.6 | 22.50 | **20.59** |
| DPO | 77.63 | 34.64 | 35.00 | 17.50 | 17.65 |
| RPO | 84.23 | 42.64 | 42.20 | 22.50 | 17.28 |
| IPO | 80.97 | 37.44 | 36.00 | 22.50 | 17.28 |
| TDPO | 82.26 | 42.32 | 42.00 | 22.50 | 19.49 |
| KTO | **84.61** | 42.82 | 43.80 | 20.00 | 17.28 |
| RePO_det | 80.44 | **46.46** | 47.40 | 22.50 | 19.85 |

## 7.2. Can RePO be Applied to Offline Datasets Without Access to the Behavior Policy?

In practice, offline preference datasets often lack explicit behavior-policy information. While **RePO** can leverage such metadata when available, it is designed to remain applicable in behavior-agnostic settings. To accommodate scenarios with restricted access to behavior policies, we introduce **RePO_det**, which approximates each trajectory as being generated by a deterministic behavior policy. This formulation preserves the core structure of **RePO** while improving practical usability, particularly in cross-model settings where the data-collecting and target policies rely on different tokenizers. We evaluate this broader applicability through experiments on the LLaMA family (Dubey et al., 2024).

As shown in Table 2, **RePO_det** achieves performance comparable to **RePO** within the Qwen model family. Moreover, the design of **RePO_det**, which does not rely on explicit access to the behavior policy, makes it amenable to training on data generated by models from different families, such as LLaMA. As shown in Table 3, **RePO_det** still achieves consistently stronger performance than existing baselines even in cross model evaluation settings. These results suggest that the sequential KL divergence term derived in **RePO** can be reliably approximated from samples, and that the resulting estimation error does not appear to dominate performance in practice. More broadly, the observed performance suggests that correctly accounting for rollout likelihoods plays an important role in effective preference based learning across heterogeneous data sources.

To further assess the generalizability of **RePO_det** across both model families and parameter scales, we additionally evaluate it on Llama3.2-1B and Gemma-2-2B, neither of

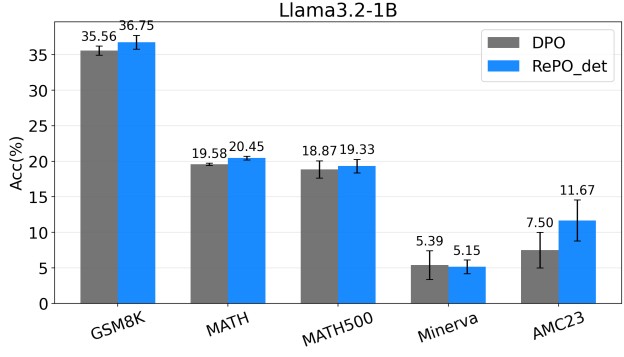

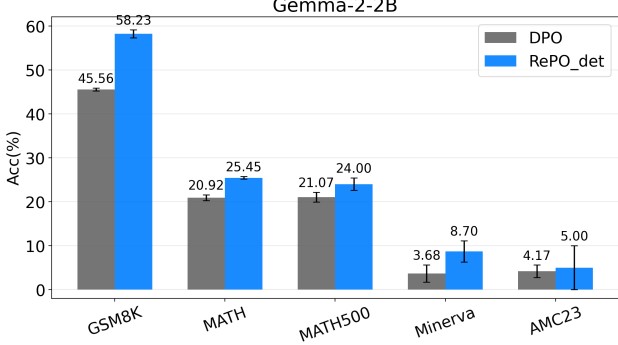

*Figure 4.* Math-reasoning accuracy of DPO and **RePO_det** on Llama3.2-1B and Gemma-2-2B, trained on preference data generated by Qwen-family models. Bars report the mean over five random seeds and error bars denote one standard deviation.

which shares a tokenizer or training pipeline with the Qwen-based data generators. Figure 4 reports the resulting math-reasoning accuracy across all five benchmarks. **RePO_det** matches or outperforms DPO on the majority of benchmarks for both backbones. These results indicate that the sample-based regret estimator remains effective and generalizes across substantial differences in model family and scale.

## 7.3. Is Regret Minimization more Sample-efficient than Reward Maximization?

As shown in Lemma 6.1, regret induces an inductive bias toward verifier-accepted outputs: when a successful trajectory is only partially observed, uncertainty about its eventual completion leads to a systematically pessimistic evaluation. We examine whether this bias is already internalized by **RePO**. To this end, we construct an augmented preference dataset by masking verifier-accepted solutions. For each preferred trajectory that reaches a verifier-accepted output, we randomly remove the final $\{16, 32, 48, 64, 80\}$ tokens, including the answer, and label the resulting context as non-preferred while retaining the original trajectory as preferred. This procedure reflects a common characteristic of human judgment, where incomplete reasoning paths are judged less favorably than fully revealed successful ones.

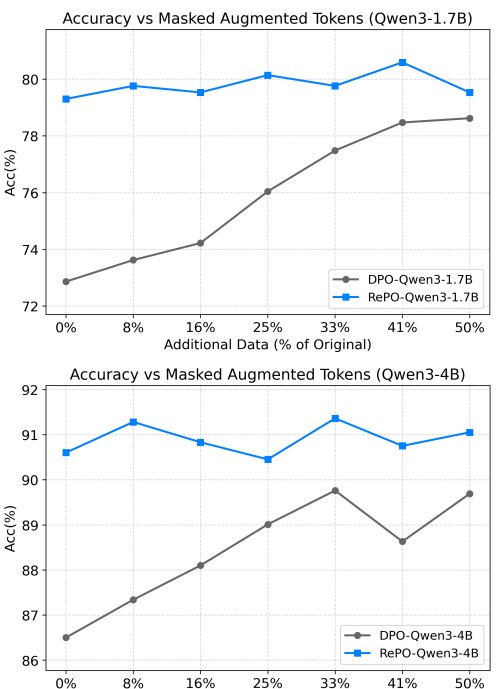

*Figure 5.* GSM8K accuracy as a function of the amount of masked-augmented preference data on Qwen3-1.7B and Qwen3-4B.

Figure 5 reports GSM8K accuracy on Qwen3-1.7B and Qwen3-4B as we vary the amount of augmented data from 0 to 30K pairs. Two patterns emerge consistently across both backbone scales. First, DPO accuracy grows roughly monotonically with the number of augmented samples, indicating that the verifier-induced inductive bias is acquired only through this additional supervision. Second, **RePO** remains nearly flat across the entire range while consistently dominating DPO, even when DPO is provided with the full 30K augmented pairs. This suggests that regret minimization structurally encodes the inductive bias that DPO must learn from data, yielding comparable or stronger performance without consuming additional supervision.

Table 4 confirms this picture under the 30K augmentation setting and provides a complementary score-level analysis. DPO benefits substantially from the augmentation, exhibiting consistent performance gains across models and benchmarks, which indicates that the injected bias provides additional supervision absent from the original objective. In contrast, **RePO** shows marginal changes, confirming that this inductive bias is already internalized by the regret-based objective. To further examine this effect, Table 4 also reports the MSE between token-level log-probability scores induced by **RePO** and those learned by each method. Notably, for DPO, training on the augmented dataset consistently reduces this MSE, indicating that data augmentation leads DPO to a scoring structure closer to that induced by **RePO**. Token-level visualizations of these log-probability differences are provided in Appendix F for qualitative interpretation.

*Table 4.* Ablation on mathematical reasoning benchmarks. Parenthesized values denote performance changes due to data augmentation (smaller is better). MSE reports the mean squared error between learned log-probability scores and RePO-induced scores.

| Model | Method | GSM8k | MATH | MSE |
|---|---|---|---|---|
| **Qwen3-1.7B** | DPO | 72.55 | 52.14 | 0.073 |
| | DPO$_{masked}$ | 77.71 (+5.16) | 53.52 (+1.38) | 0.056 |
| | RePO | 80.52 | 54.50 | 0 |
| | RePO$_{masked}$ | 80.29 (-0.23) | 55.26 (+0.76) | - |
| **Qwen3-4B** | DPO | 85.22 | 63.70 | 0.066 |
| | DPO$_{masked}$ | 88.63 (+3.41) | 64.46 (+0.72) | 0.054 |
| | RePO | 90.60 | 65.54 | 0 |
| | RePO$_{masked}$ | 90.98 (+0.38) | 65.46 (-0.08) | - |

## 8. Conclusions

We introduced Regret-based Preference Optimization (**RePO**), which reframes preference learning for LLMs through regret minimization rather than reward maximization. By modeling human feedback as inherently prospective and counterfactual, **RePO** provides a principled interpretation of preferences over heterogeneous trajectories. We derived a closed-form regret decomposition under KL-regularized RL and demonstrated improvements over DPO-style baselines on human preference and reasoning benchmarks. In conclusion, our results suggest that regret offers a faithful abstraction of human feedback and a promising foundation for human-aligned post-training of LLMs.

## Acknowledgements

This work is in part supported by the National Research Foundation of Korea (NRF, RS-2026-25486774(10%), RS-2026-25474295(10%) RS-2024-00451435(10%), RS-2024-00413957(10%)), Institute of Information & communications Technology Planning & Evaluation (IITP, RS-2026-25523906(10%), Hyper-scale Industrial AI Research Support (R&D) Program, Development of an industry-specified multimodal hyperscale foundation model, RS-2025-02305453(10%), RS-2025-02273157(10%), RS-2025-25442149(10%), RS-2021-II211343(10%), RS-2020-II201336(10%) ) grant funded by the Ministry of Science and ICT (MSIT), Institute of New Media and Communications(INMAC), and the BK21 FOUR program of the Education, Artificial Intelligence Graduate School Program (Seoul National University, UNIST), and Research Program for Future ICT Pioneers, Seoul National University in 2026.

## Impact Statement

This paper presents work whose goal is to advance the field of Machine Learning. There are many potential societal consequences of our work, none which we feel must be specifically highlighted here.

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

# A. Main Proofs

This section provides proofs for the theoretical results presented in the main text. The proof techniques used for Lemma 5.3, Lemma A.1, and Theorem 5.4 are adapted from Cho et al. (2025) and extend prior analyses from maximum entropy reinforcement learning to KL-regularized RL framework.

**Lemma** (5.1). *Let the Q-function of policy $\mu$ be defined as*

$$Q^\mu(q_{<t}, o) := r(q_{<t}, o) + \mathbb{E}_{\mathbb{P}^\mu}\Big[\sum_{l>0}\gamma^l\big(r(q_{t+l}, o_{t+l}) - \alpha\mathbb{D}_{KL}(\mu(\cdot|q_{<t+l})||\pi_{ref}(\cdot|q_{<t+l})))\big)\Big].$$

*Define value function $V^\pi$ satisfying the Bellman equation*

$$V^\mu(q_{<t}) = \mathbb{E}_\mu[Q^\pi(q_{<t}, o) - \alpha\mathbb{D}_{KL}(\mu(\cdot|q_{<t})||\pi_{ref}(\cdot|q_{<t}))]$$
$$Q^\mu(q_{<t}, o) = r(q_{<t}, o) + \gamma\mathbb{E}_{q_{<t+1}\sim\mathbb{P}^\mu(\cdot|q_{<t},o)}[V^\mu(q_{<t+1})].$$

*Then the optimal value function and the optimal policy satisfies the following relation:*

$$V^{\pi^*}(q_{<t}) = \alpha\log\mathbb{E}_{o\sim\pi_{ref}}\Big[\exp\Big(\frac{1}{\alpha}Q^{\pi^*}(q_{<t}, o)\Big)\Big],$$
$$\frac{\pi^*(o|q_{<t})}{\pi_{ref}(o|q_{<t})} = \exp\Big(\frac{1}{\alpha}(Q^{\pi^*}(q_{<t}, o) - V^{\pi^*}(q_{<t}))\Big).$$

*Proof.* We first establish a policy-improvement argument for the KL-regularized objective. Given a policy $\pi_k$, define $\pi_{k+1}$ pointwise by

$$\forall q_{<t}, \quad \pi_{k+1}(\cdot|q_{<t}) \in \arg\max_\pi\Big\{\mathbb{E}_\pi\big[Q^{\pi_k}(q_{<t}, o)\big] - \alpha\,\mathbb{D}_{KL}\big(\pi(\cdot|q_{<t})||\pi_{ref}(\cdot|q_{<t})\big)\Big\}. \tag{3}$$

Since the objective in (3) is concave in $\pi$, the maximizer satisfies, for all $q_{<t}$,

$$\mathbb{E}_{\pi_{k+1}}\big[Q^{\pi_k}(q_{<t}, o)\big] - \alpha\,\mathbb{D}_{KL}\big(\pi_{k+1}(\cdot|q_{<t})||\pi_{ref}(\cdot|q_{<t})\big) \geq \mathbb{E}_{\pi_k}\big[Q^{\pi_k}(q_{<t}, o)\big] - \alpha\,\mathbb{D}_{KL}\big(\pi_k(\cdot|q_{<t})||\pi_{ref}(\cdot|q_{<t})\big), \tag{4}$$

with equality iff $\pi_{k+1} = \pi_k$.

Next, using (4) and the Bellman equation for $Q^{\pi_k}$, we obtain the standard monotonicity:

$$\begin{aligned}
Q^{\pi_k}(q_{<t}, o) &= r(q_{<t}, o) + \gamma\,\mathbb{E}_{q_{<t+1}\sim\mathbb{P}(\cdot|q_{<t},o)}\big[V^{\pi_k}(q_{<t+1})\big] \\
&\leq r(q_{<t}, o) + \gamma\,\mathbb{E}_{q_{<t+1}\sim\mathbb{P}(\cdot|q_{<t},o)}\Big[\mathbb{E}_{\pi_{k+1}}\big[Q^{\pi_k}(q_{<t+1}, \cdot)\big] - \alpha\,\mathbb{D}_{KL}\big(\pi_{k+1}(\cdot|q_{<t+1})||\pi_{ref}(\cdot|q_{<t+1})\big)\Big] \\
&\;\;\vdots \\
&\leq r(q_{<t}, o) + \mathbb{E}_{\mathbb{P}^{\pi_{k+1}}}\Big[\sum_{l>0}\gamma^l\big(r(q_{<t+l}, o_{t+l}) - \alpha\,\mathbb{D}_{KL}\big(\pi_{k+1}(\cdot|q_{<t+l})||\pi_{ref}(\cdot|q_{<t+l})\big)\big)\Big] \\
&= Q^{\pi_{k+1}}(q_{<t}, o),
\end{aligned}$$

where the second inequality follows by iterating the one-step bound and using $\gamma^{l+1}V^{\pi_k}(q_{<t+l+1}) \to 0$ as $l \to \infty$. Therefore, $\{Q^{\pi_k}\}_{k\geq 0}$ is pointwise non-decreasing and bounded, and hence converges to some $\tilde{\pi}$. By construction in (3), $\tilde{\pi}$ is a fixed point of the improvement step and is thus optimal.

It remains to derive the closed-form expressions. The maximization in (3) is solved by Lagrangian optimization, yielding

$$\forall q_{<t}, \quad \pi_{k+1}(o|q_{<t}) = \frac{\pi_{ref}(o|q_{<t})\exp\big(\frac{1}{\alpha}Q^{\pi_k}(q_{<t}, o)\big)}{Z^{\pi_k}(q_{<t})}, \qquad Z^{\pi_k}(q_{<t}) := \mathbb{E}_{o\sim\pi_{ref}(\cdot|q_{<t})}\Big[\exp\big(\frac{1}{\alpha}Q^{\pi_k}(q_{<t}, o)\big)\Big]. \tag{5}$$

At optimality, setting $\pi_{k+1} = \pi_k = \pi^*$ in (5) gives

$$\pi^*(o|q_{<t}) = \frac{\pi_{ref}(o|q_{<t})\exp\big(\frac{1}{\alpha}Q^{\pi^*}(q_{<t}, o)\big)}{Z^{\pi^*}(q_{<t})}. \tag{6}$$

Taking logs in (6) yields

$$\alpha \log \frac{\pi^*(\cdot|q_{<t})}{\pi_{\text{ref}}(\cdot|q_{<t})} = Q^{\pi^*}(q_{<t}, \cdot) - \alpha \log Z^{\pi^*}(q_{<t}).$$

Finally, by the definition of the KL-regularized value,

$$V^{\pi^*}(q_{<t}) = \mathbb{E}_{\pi^*}\big[Q^{\pi^*}(q_{<t}, \cdot)\big] - \alpha \, \mathbb{D}_{\text{KL}}\big(\pi^*(\cdot|q_{<t})\|\pi_{\text{ref}}(\cdot|q_{<t})\big)$$

$$= \alpha \log Z^{\pi^*}(q_{<t}) = \alpha \log \mathbb{E}_{o \sim \pi_{\text{ref}}(\cdot|q_{<t})}\Big[\exp\big(\tfrac{1}{\alpha}Q^{\pi^*}(q_{<t}, o)\big)\Big],$$

which completes the proof. ∎

**Lemma** (5.3). *(Structural Condition for $\alpha$-optimality) An optimal $Q$-function where $\pi^*(\cdot|q_{<t})$ is $\alpha$-optimal have a one-to-one correspondence with a state-dependent function $\beta : \mathcal{S} \to \mathbb{R}$, defined as follows:*

$$\mathcal{R}^{\pi^*} = \left\{ R^{\pi^*}(q_{<t}, o) = \alpha \log \frac{\pi^*(o|q_{<t})}{\pi_{ref}(o|q_{<t})} + \beta(q_{<t}) - \mathbb{E}_{\mathbb{P}}[\beta(q_{<t+1})] \right\},$$

$$\mathcal{Q}^{\pi^*} = \left\{ Q^{\pi^*}(q_{<t}, o) = \alpha \log \frac{\pi^*(o|q_{<t})}{\pi_{ref}(o|q_{<t})} + \beta(q_{<t}) \right\}.$$

*Proof.* According to Lemma 5.1, remark that the policy $\pi^*$ is $\alpha$-optimal, if and only if there exists the $Q$-function satisfies the following relation:

$$\pi^*(o|q_{<t}) \propto \pi_{\text{ref}}(o|q_{<t}) \exp\left(\frac{1}{\alpha}Q^{\pi^*}(q_{<t}, o)\right), \quad V^{\pi^*}(q_{<t}) = \alpha \log \mathbb{E}_{\pi_{\text{ref}}}\left[\exp(\frac{1}{\alpha}Q^{\pi^*}(q_{<t}, o))\right].$$

Since $V^{\pi^*}$ is merely a partition function, letting $X(q_{<t}, o) = \pi_{\text{ref}}(o|q_{<t}) \exp\left(\frac{1}{\alpha}Q^{\pi^*}(q_{<t}, o)\right)$, we can derive

$$\pi^*(o|q_{<t}) = \frac{X(q_{<t}, o)}{\sum_{o \in \mathcal{O}} X(q_{<t}, o)} \iff X(q_{<t}, o) = d(q_{<t})\pi^*(o|q_{<t}) \text{ for some } d : \mathcal{S} \to \mathbb{R}$$

$$\iff Q^{\pi^*}(q_{<t}, o) = \alpha \log \frac{\pi^*(o|q_{<t})}{\pi_{\text{ref}}(o|q_{<t})} + \beta(q_{<t}) \text{ for some } \beta : \mathcal{S} \to \mathbb{R},$$

where $\beta$ is defined as $\beta(q_{<t}) = \alpha \log d(q_{<t})$.

Recall the definition of $Q$ and $V$-function of KL-divergence RL framework:

$$Q^{\pi^*}(q_{<t}, o_t) = r(q_{<t}, o_t) + \gamma \mathbb{E}_{\mathbb{P}}[V^{\pi^*}(q_{<t+1})],$$

$$V^{\pi^*}(q_{<t}) := \max_{\mu(\cdot|s)} \mathbb{E}_{\mu}\Big[Q^{\mu}(\cdot|s) - \alpha\mathbb{D}_{\text{KL}}(\mu(\cdot|s)\|\pi_{\text{ref}}(\cdot|s))\Big].$$

Then, the reward function $r(q_{<t}, o_t)$ can be expressed as:

$$r(q_{<t}, o_t) = Q^{\pi^*}(q_{<t}, o_t) - \gamma \mathbb{E}_{\mathbb{P}}[V^{\pi^*}(q_{<t+1})]$$

$$= \alpha \log \frac{\pi^*(o_t|q_{<t})}{\pi_{\text{ref}}(o_t|q_{<t})} + \beta(q_{<t}) - \gamma\mathbb{E}_{\mathbb{P}, \pi^*}\left[\alpha \log \frac{\cancel{\pi^*(o|q_{<t+1})}}{\cancel{\pi_{\text{ref}}(o|q_{<t+1})}} + \beta(q_{<t+1}) \cancel{-\alpha\mathbb{D}_{\text{KL}}(\mu(\cdot|q_{<t+1})\|\pi_{\text{ref}}(\cdot|q_{<t+1}))}\right]$$

$$= \alpha \log \frac{\pi^*(o_t|q_{<t})}{\pi_{\text{ref}}(o_t|q_{<t})} + \beta(q_{<t}) - \gamma\mathbb{E}_{\mathbb{P}, \pi^*}[\beta(q_{<t+1})]$$

∎

**Lemma A.1** (Unique Fixed Point of Bellman $\mu$-operator (KL-regularized)). *Let $\pi^*$ be $\alpha$-optimal under the KL-regularized objective with reference policy $\pi_{\text{ref}}$. Fix an $\alpha$-optimal reward $r_*$ (equivalently, an optimal $Q$-function $Q^{\pi^*}_*$). For a given policy $\mu$ and any $Q^{\mu}_A \in \mathcal{Q}^{\mu}$, define the Bellman $\mu$-operator $\mathcal{T}^{\mu}_* : \mathcal{Q}^{\mu} \to \mathcal{Q}^{\mu}$ by, for all $(q_{<t}, o)$,*

$$\mathcal{T}^{\mu}_* Q^{\mu}_A(q_{<t}, o_t)$$

$$:= Q^{\pi^*}_*(q_{<t}, o_t) - \gamma \, \mathbb{E}_{q_{<t+1} \sim \mathbb{P}(\cdot|q_{<t}, o_t)}\Big[\alpha\Big(\mathbb{D}_{KL}\big(\mu(\cdot|q_{<t+1})\|\pi_{\text{ref}}(\cdot|q_{<t+1})\big) - \mathbb{D}_{KL}\big(\pi^*(\cdot|q_{<t+1})\|\pi_{\text{ref}}(\cdot|q_{<t+1})\big)\Big)$$

$$+ \mathbb{E}_{o \sim \pi^*(\cdot|q_{<t+1})}\big[Q^{\pi^*}_*(q_{<t+1}, o)\big] - \mathbb{E}_{o \sim \mu(\cdot|q_{<t+1})}\big[Q^{\mu}_A(q_{<t+1}, o)\big]\Big].$$

*Then, $\mathcal{T}^{\mu}_*$ has a unique fixed point, denoted by $Q^{\mu}_*$.*

*Proof.* Consider any $Q_A^\mu, Q_B^\mu \in \mathcal{Q}^\mu$. Since the terms involving $Q_*^{\pi^*}$ and the KL-divergences do not depend on $Q_A^\mu$ or $Q_B^\mu$, we have

$$\sup_{q_{<t}, o_t} \left| \mathcal{T}_*^\mu Q_A^\mu(q_{<t}, o_t) - \mathcal{T}_*^\mu Q_B^\mu(q_{<t}, o_t) \right| = \sup_{q_{<t}, o_t} \left| \gamma \, \mathbb{E}_{q_{<t+1} \sim \mathbb{P}(\cdot|q_{<t}, o_t)} \left[ \mathbb{E}_{o \sim \mu(\cdot|q_{<t+1})} \left[ Q_A^\mu(q_{<t+1}, o) - Q_B^\mu(q_{<t+1}, o) \right] \right] \right|$$

$$\leq \gamma \sup_{q_{<t+1}, o} \left| Q_A^\mu(q_{<t+1}, o) - Q_B^\mu(q_{<t+1}, o) \right|.$$

Hence, $\mathcal{T}_*^\mu$ is a $\gamma$-contraction under the sup norm. Since $\mathcal{Q}^\mu$ is complete, the Banach fixed-point theorem implies that $\mathcal{T}_*^\mu$ admits a unique fixed point.

It remains to verify that $Q_*^\mu$ is indeed a fixed point. Under KL-regularized RL, the Bellman equations for $\pi^*$ and $\mu$ (under the same $r_*$) are

$$Q_*^{\pi^*}(q_{<t}, o_t)$$
$$= \mathbb{E}_{q_{<t+1} \sim \mathbb{P}(\cdot|q_{<t}, o_t)} \left[ r_*(q_{<t}, o_t) + \gamma \left( \mathbb{E}_{o \sim \pi^*(\cdot|q_{<t+1})} [Q_*^{\pi^*}(q_{<t+1}, o)] - \alpha \, \mathbb{D}_{\mathrm{KL}} \left( \pi^*(\cdot|q_{<t+1}) \| \pi_{\mathrm{ref}}(\cdot|q_{<t+1}) \right) \right) \right],$$
$$Q_*^\mu(q_{<t}, o_t)$$
$$= \mathbb{E}_{q_{<t+1} \sim \mathbb{P}(\cdot|q_{<t}, o_t)} \left[ r_*(q_{<t}, o_t) + \gamma \left( \mathbb{E}_{o \sim \mu(\cdot|q_{<t+1})} [Q_*^\mu(q_{<t+1}, o)] - \alpha \, \mathbb{D}_{\mathrm{KL}} \left( \mu(\cdot|q_{<t+1}) \| \pi_{\mathrm{ref}}(\cdot|q_{<t+1}) \right) \right) \right].$$

Using the first equation to rewrite $Q_*^{\pi^*}(q_{<t}, o_t)$ and substituting into the operator,

$$\mathcal{T}_*^\mu Q_*^\mu(q_{<t}, o_t)$$
$$= Q_*^{\pi^*}(q_{<t}, o_t) - \gamma \mathbb{E}_{q_{<t+1} \sim \mathbb{P}(\cdot|q_{<t}, o_t)} \left[ \alpha \left( \mathbb{D}_{\mathrm{KL}} \left( \mu(\cdot|q_{<t+1}) \| \pi_{\mathrm{ref}}(\cdot|q_{<t+1}) \right) - \mathbb{D}_{\mathrm{KL}} \left( \pi^*(\cdot|q_{<t+1}) \| \pi_{\mathrm{ref}}(\cdot|q_{<t+1}) \right) \right) \right.$$
$$\left. + \mathbb{E}_{o \sim \pi^*(\cdot|q_{<t+1})} [Q_*^{\pi^*}(q_{<t+1}, o)] - \mathbb{E}_{o \sim \mu(\cdot|q_{<t+1})} [Q_A^\mu(q_{<t+1}, o)] \right]$$
$$= \mathbb{E}_{q_{<t+1} \sim \mathbb{P}(\cdot|q_{<t}, o_t)} \left[ r_*(q_{<t}, o_t) + \gamma \left( \mathbb{E}_{\pi^*} [Q_*^{\pi^*}(q_{<t+1}, \cdot)] - \alpha \mathbb{D}_{\mathrm{KL}}(\pi^*(\cdot|q_{<t+1}) \| \pi_{\mathrm{ref}}(\cdot|q_{<t+1})) \right) \right]$$
$$- \gamma \mathbb{E}_{q_{<t+1} \sim \mathbb{P}(\cdot|q_{<t}, o_t)} \left[ \alpha \mathbb{D}_{\mathrm{KL}}(\pi(\cdot|q_{<t+1}) \| \pi_{\mathrm{ref}}(\cdot|q_{<t+1})) - \alpha \mathbb{D}_{\mathrm{KL}}(\pi^*(\cdot|q_{<t+1}) \| \pi_{\mathrm{ref}}(\cdot|q_{<t+1})) \right.$$
$$\left. + \mathbb{E}_{\pi^*} [Q_*^{\pi^*}(q_{<t+1}, \cdot)] - \mathbb{E}_\mu [Q_*^\mu(q_{<t+1}, \cdot)] \right]$$
$$= \mathbb{E}_{q_{<t+1} \sim \mathbb{P}(\cdot|q_{<t}, o_t)} \left[ r_*(q_{<t}, o_t) + \gamma \left( \mathbb{E}_\mu [Q_*^\mu(q_{<t+1}, \cdot)] - \alpha \mathbb{D}_{\mathrm{KL}}(\pi(\cdot|q_{<t+1}) \| \pi_{\mathrm{ref}}(\cdot|q_{<t+1})) \right) \right]$$
$$= Q_*^\mu(q_{<t}, o_t),$$

which shows that $Q_*^\mu$ is a fixed point. By uniqueness of the fixed point, $Q_*^\mu$ is the unique fixed point of $\mathcal{T}_*^\mu$. ∎

**Theorem (5.4).** *(KL-divergence Policy Deviation Theorem) If a policy $\pi^*$ is $\alpha$-optimal, then for any policy $\mu$,*

$$Q_*^{\pi^*}(q_{<t}, o) - Q_*^\mu(q_{<t}, o) = \alpha \bar{\mathbb{D}}_{KL}(\mu \| \pi^*; q_{<t}, o),$$

*where the **sequential forward KL-divergence** is defined as*

$$\bar{\mathbb{D}}_{KL}(\mu \| \pi^*; q_{<t}, o) := \mathbb{E}_{\tau \sim \mathbb{P}^\mu} \left[ \sum_{l>0} \gamma^l \mathbb{D}_{KL} \left( \mu(\cdot|q_{<t+l}) \| \pi^*(\cdot|q_{<t+l}) \right) \right].$$

*Here, $\mathbb{P}^\mu$ denotes the distribution of trajectories induced by policy $\mu$ and transition kernel $\mathbb{P}$, starting from the initial context-action pair $(q_{<t}, o)$.*

*Proof.* For all $(q_{<t}, o_t)$, define

$$\widetilde{Q}_*^\mu(q_{<t}, o_t) := Q_*^{\pi^*}(q_{<t}, o_t) - \alpha \bar{\mathbb{D}}_{\mathrm{KL}}(\mu \| \pi^*; q_{<t}, o_t).$$

We will show that $\widetilde{Q}_*^\mu$ is a fixed point of the Bellman $\mu$-operator $\mathcal{T}_*^\mu$ in Lemma A.1. By uniqueness of the fixed point, this will imply $\widetilde{Q}_*^\mu = Q_*^\mu$ and hence the desired identity.

First, recall from Lemma 5.3 that there exists a state-dependent function $\beta(\cdot)$ such that, for all $(q_{<t}, o)$,

$$Q_*^{\pi^*}(q_{<t}, o) = \alpha \log \frac{\pi^*(o \mid q_{<t})}{\pi_{\text{ref}}(o \mid q_{<t})} + \beta(q_{<t}). \tag{7}$$

Taking expectation with respect to a policy $\nu(\cdot|q_{<t})$ and using $\mathbb{D}_{\text{KL}}(\nu\|\pi_{\text{ref}}) = \mathbb{E}_\nu\left[\log \frac{\nu}{\pi_{\text{ref}}}\right]$, we obtain

$$\mathbb{E}_{o\sim\nu(\cdot|q_{<t})}[Q_*^{\pi^*}(q_{<t}, o)] - \alpha\,\mathbb{D}_{\text{KL}}\big(\nu(\cdot|q_{<t})\|\pi_{\text{ref}}(\cdot|q_{<t})\big) = \beta(q_{<t}) - \alpha\,\mathbb{D}_{\text{KL}}\big(\nu(\cdot|q_{<t})\|\pi^*(\cdot|q_{<t})\big). \tag{8}$$

In particular, (8) yields

$$\mathbb{E}_{\pi^*}\big[Q_*^{\pi^*}(q_{<t}, \cdot)\big] - \alpha\,\mathbb{D}_{\text{KL}}\big(\pi^*(\cdot|q_{<t})\|\pi_{\text{ref}}(\cdot|q_{<t})\big) = \beta(q_{<t}), \tag{9}$$

$$\mathbb{E}_{\mu}\big[Q_*^{\pi^*}(q_{<t}, \cdot)\big] - \alpha\,\mathbb{D}_{\text{KL}}\big(\mu(\cdot|q_{<t})\|\pi_{\text{ref}}(\cdot|q_{<t})\big) = \beta(q_{<t}) - \alpha\,\mathbb{D}_{\text{KL}}\big(\mu(\cdot|q_{<t})\|\pi^*(\cdot|q_{<t})\big). \tag{10}$$

Now apply the operator $\mathcal{T}_*^\mu$ to $\widetilde{Q}_*^\mu$:

$$\mathcal{T}_*^\mu\widetilde{Q}_*^\mu(q_{<t}, o_t) = Q_*^{\pi^*}(q_{<t}, o_t) - \gamma\,\mathbb{E}_{q_{<t+1}}\Big[\alpha\Big(\mathbb{D}_{\text{KL}}(\mu(\cdot|q_{<t+1})\|\pi_{\text{ref}}(\cdot|q_{<t+1})) - \mathbb{D}_{\text{KL}}(\pi^*(\cdot|q_{<t+1})\|\pi_{\text{ref}}(\cdot|q_{<t+1}))\Big)$$
$$+ \mathbb{E}_{\pi^*}\big[Q_*^{\pi^*}(q_{<t+1}, \cdot)\big] - \mathbb{E}_\mu\big[\widetilde{Q}_*^\mu(q_{<t+1}, \cdot)\big]\Big].$$

Using $\widetilde{Q}_*^\mu = Q_*^{\pi^*} - \alpha\,\bar{\mathbb{D}}_{\text{KL}}(\mu\|\pi^*; \cdot)$, we rewrite the last expectation:

$$\mathbb{E}_\mu\big[\widetilde{Q}_*^\mu(q_{<t+1}, \cdot)\big] = \mathbb{E}_\mu\big[Q_*^{\pi^*}(q_{<t+1}, \cdot)\big] - \alpha\,\mathbb{E}_\mu\big[\bar{\mathbb{D}}_{\text{KL}}(\mu\|\pi^*; q_{<t+1}, \cdot)\big].$$

Substituting this and regrouping terms gives

$$\mathcal{T}_*^\mu\widetilde{Q}_*^\mu(q_{<t}, o_t)$$
$$= Q_*^{\pi^*}(q_{<t}, o_t) - \gamma\,\mathbb{E}_{q_{<t+1}}\Big[\big(\mathbb{E}_{\pi^*}[Q_*^{\pi^*}(q_{<t+1}, \cdot)] - \alpha\mathbb{D}_{\text{KL}}(\pi^*(\cdot|q_{<t+1})\|\pi_{\text{ref}}(\cdot|q_{<t+1}))\big)$$
$$- \big(\mathbb{E}_\mu[Q_*^{\pi^*}(q_{<t+1}, \cdot)] - \alpha\mathbb{D}_{\text{KL}}(\mu(\cdot|q_{<t+1})\|\pi_{\text{ref}}(\cdot|q_{<t+1}))\big) + \alpha\,\mathbb{E}_\mu\big[\bar{\mathbb{D}}_{\text{KL}}(\mu\|\pi^*; q_{<t+1}, \cdot)\big]\Big].$$

Applying (9) and (10) at $q_{<t+1}$, the difference in parentheses becomes

$$\big(\mathbb{E}_{\pi^*}[Q_*^{\pi^*}(q_{<t+1}, \cdot)] - \alpha\mathbb{D}_{\text{KL}}(\pi^*(\cdot|q_{<t+1})\|\pi_{\text{ref}}(\cdot|q_{<t+1})))\big)$$
$$- \big(\mathbb{E}_\mu[Q_*^{\pi^*}(q_{<t+1}, \cdot)] - \alpha\mathbb{D}_{\text{KL}}(\mu(\cdot|q_{<t+1})\|\pi_{\text{ref}}(\cdot|q_{<t+1})))\big) = \alpha\,\mathbb{D}_{\text{KL}}\big(\mu(\cdot|q_{<t+1})\|\pi^*(\cdot|q_{<t+1})\big).$$

Therefore,

$$\mathcal{T}_*^\mu\widetilde{Q}_*^\mu(q_{<t}, o_t) = Q_*^{\pi^*}(q_{<t}, o_t) - \alpha\gamma\,\mathbb{E}_{q_{<t+1}}\Big[\mathbb{D}_{\text{KL}}\big(\mu(\cdot|q_{<t+1})\|\pi^*(\cdot|q_{<t+1})\big) + \mathbb{E}_\mu\big[\bar{\mathbb{D}}_{\text{KL}}(\mu\|\pi^*; q_{<t+1}, \cdot)\big]\Big].$$

By the definition of $\bar{\mathbb{D}}_{\text{KL}}$, the bracketed term is exactly the one-step recursion:

$$\bar{\mathbb{D}}_{\text{KL}}(\mu\|\pi^*; q_{<t}, o_t) = \gamma\,\mathbb{E}_{q_{<t+1}}\Big[\mathbb{D}_{\text{KL}}\big(\mu(\cdot|q_{<t+1})\|\pi^*(\cdot|q_{<t+1})\big) + \mathbb{E}_\mu\big[\bar{\mathbb{D}}_{\text{KL}}(\mu\|\pi^*; q_{<t+1}, \cdot)\big]\Big].$$

Hence,

$$\mathcal{T}_*^\mu\widetilde{Q}_*^\mu(q_{<t}, o_t) = Q_*^{\pi^*}(q_{<t}, o_t) - \alpha\,\bar{\mathbb{D}}_{\text{KL}}(\mu\|\pi^*; q_{<t}, o_t) = \widetilde{Q}_*^\mu(q_{<t}, o_t),$$

so $\widetilde{Q}_*^\mu$ is a fixed point of $\mathcal{T}_*^\mu$. By Lemma A.1, $\mathcal{T}_*^\mu$ has a unique fixed point $Q_*^\mu$, therefore $\widetilde{Q}_*^\mu = Q_*^\mu$, i.e.,

$$Q_*^\mu(q_{<t}, o) = Q_*^{\pi^*}(q_{<t}, o) - \alpha\,\bar{\mathbb{D}}_{\text{KL}}(\mu\|\pi^*; q_{<t}, o),$$

which proves the theorem. $\blacksquare$

**Lemma** (6.1). *Assume* $\epsilon \geq 0$ *where* $\epsilon := \mathbb{D}_{KL}(\mu(\cdot|q_{<t})\|\pi_{ref}(\cdot|q_{<t})) - \mathbb{D}_{KL}(\mu(\cdot|q_{<t})\|\pi^*(\cdot|q_{<t}))$. *Then, for any verifier-accepted output* $o_T^\star$,

$$\widehat{Reg}_{\pi^*}^\mu(q_{<T}, o_T^\star) \leq \mathbb{E}_\mu[\widehat{Reg}_{\pi^*}^\mu(q_{<t}, \cdot)].$$

*Proof.*

$$\text{Reg}^{\mu}_{\pi^*}(q_{<T}, o_T^\star) = V^{\pi^*}(q_{<T}) - Q^{\mu}(q_{<T}, o_T^\star)$$

$$= -\alpha \log \frac{\pi^*(o_T^\star | q_{<T})}{\pi_{\text{ref}}(o_T^\star | q_{<T})}$$

$$\leq 0 \qquad\qquad (\because \pi^*(o_T^\star | q) \geq \pi_{\text{ref}}(o_T^\star | q))$$

and for any $t \leq T$,

$$\mathbb{E}_\mu[\text{Reg}^{\mu}_{\pi^*}(q_{<t}, \cdot)] = \mathbb{E}_\mu[V^{\pi^*}(q_{<t}) - Q^{\mu}(q_{<t}, \cdot)]$$

$$= \mathbb{E}_\mu[-\alpha \log \frac{\pi^*(\cdot | q_{<t})}{\pi_{\text{ref}}(\cdot | q_{<t})} + \alpha \bar{\mathbb{D}}_{\text{KL}}(\mu || \pi^*; q_{<t}, o_t)]$$

$$= \mathbb{E}_\mu[\alpha \log \frac{\mu(\cdot | q_{<t})}{\pi^*(\cdot | q_{<t})} - \alpha \log \frac{\mu(\cdot | q_{<t})}{\pi_{\text{ref}}(\cdot | q_{<t})}] + \alpha \mathbb{E}_\mu[\bar{\mathbb{D}}_{\text{KL}}(\mu || \pi^*; q_{<t}, \cdot)]$$

$$= \underbrace{\alpha \mathbb{D}_{\text{KL}}(\mu(\cdot | q_{<t}) || \pi^*(\cdot | q_{<t})) - \alpha \mathbb{D}_{\text{KL}}(\mu(\cdot | q_{<t}) || \pi_{\text{ref}}(\cdot | q_{<t}))}_{\alpha\epsilon} + \alpha \mathbb{E}_\mu[\bar{\mathbb{D}}_{\text{KL}}(\mu || \pi^*; q_{<t}, \cdot)]$$

$$\geq 0$$

Therefore, $\mathbb{E}_\mu[\text{Reg}^{\mu}_{\pi^*}(q_{<t}, \cdot)] \geq \text{Reg}^{\mu}_{\pi^*}(q_{<T}, o_T^\star)$. $\blacksquare$

## B. Comparison with Policy-labeled Preference Learning (Cho et al., 2025)

This work is closely related to Policy-Labeled Preference Learning (PPL; Cho et al. (2025)) and draws substantial inspiration from its regret-based formulation. Nevertheless, our approach differs from PPL in several important aspects, both theoretically and empirically, particularly in the context of large language models.

**Different optimization frameworks.** While Cho et al. (2025) develops regret minimization under a maximum-entropy reinforcement learning objective, this paper formulates regret within a KL-regularized RL framework that is standard in language-model post-training. Under this transition, the local preference term changes from $\alpha \log \pi^*$ to $\alpha \log \frac{\pi^*}{\pi_{\text{ref}}}$, reflecting the explicit use of a reference policy intrinsic to KL-regularized objectives. Importantly, the sequential forward KL divergence term, which captures long-horizon policy deviation, remains unchanged. This demonstrates that the core regret structure is preserved while adapting the theory to a setting that more faithfully reflects practical LLM training pipelines.

**Human-centered justification of regret.** This paper provides a cognitive justification for regret minimization grounded in prospective and counterfactual human reasoning. Drawing on insights from cognitive psychology, we explain why humans can meaningfully express preferences over intermediate or partial trajectories even when no verifier-accessible outcome is available. This perspective naturally supports learning from preference feedback on intermediate reasoning steps, which commonly arises in LLM settings but is difficult to justify under reward-maximization alone.

In contrast, Cho et al. (2025) motivates regret primarily as a mechanism to prevent likelihood mismatch caused by MDP indistinguishability in stochastic environments, where suboptimal learning may arise from confounding between policy and transition dynamics. While this justification is appropriate for stochastic control settings, it becomes less direct in deterministic environments such as LLM reasoning, where the next state is defined as a deterministic concatenation of the current context and output. In such settings, likelihood mismatch due to transition uncertainty does not arise. Our work addresses this gap by showing that regret remains a principled and human-aligned learning objective even in fully deterministic environments.

**Inductive bias of regret-based human feedback.** Finally, Lemma 6.1 reveals that the regret minimization framework structurally internalizes an inductive bias present in human feedback. In particular, partially observed or masked trajectories are evaluated pessimistically relative to fully revealed ones. Empirically, RePO does not require additional data augmentation to enforce such preference relations, yet achieves strong performance. This demonstrates that regret-based preference optimization can be more sample-efficient by internalizing structural properties of human feedback rather than relying on externally imposed heuristics.

# C. Reward Anchoring Schemes and the Role of $\beta(\cdot)$

In this appendix, we provide explicit formulations of commonly used reward-anchoring schemes that mitigate the ambiguity induced by the state-dependent shaping function $\beta(\cdot)$ identified in Lemma 5.3. These approaches aim to fix an absolute reward scale under KL-regularized reward maximization, thereby reducing sensitivity to reward translations under finite data.

**In-distribution Reward Normalization**  A common approach to mitigating the ambiguity induced by the shaping function $\beta(\cdot)$ is to normalize rewards using in-distribution statistics. Recent RLHF methods such as REINFORCE Leave-One-Out (`RLOO`) (Ahmadian et al., 2024) and Group Relative Policy Optimization (`GRPO`) (Guo et al., 2025) provide concrete instances of this strategy, by standardizing reward or advantage signals using empirical mean and variance computed from sampled data. Concretely, given an empirical distribution $\mathcal{D}$ of in-distribution context–action pairs, RLOO normalizes rewards by removing their empirical mean and scaling by their variance:

$$\widehat{r}_{\text{RLOO}}(q_{<t}, o) = r(q_{<t}, o) - \mathbb{E}_{\mathcal{D} \setminus \{o\}}[r]. \tag{11}$$

Similarly, GRPO applies normalization at the level of groups sharing the same prompt or context,

$$\widehat{r}_{\text{GRPO}}(q_{<t}, o) = \frac{r(q_{<t}, o) - \mathbb{E}_{\mathcal{G}}[r]}{\sqrt{\text{Var}_{\mathcal{G}}[r]}}, \tag{12}$$

where $\mathcal{G}$ denotes a group of samples generated from the same input.

From the perspective of Lemma 5.3, these operations admit a clear interpretation. Subtracting the empirical mean corresponds to fixing the additive shaping function $\beta(\cdot)$ up to a constant, thereby anchoring the reward scale on the data distribution. In contrast, dividing by the empirical standard deviation amounts to rescaling the effective temperature parameter $\alpha$. Specifically, replacing $r$ with $(r - \mathbb{E}[r])/\sqrt{\text{Var}[r]}$ is equivalent to modifying the KL-regularized objective with a rescaled temperature $\alpha' = \alpha/\sqrt{\text{Var}[r]}$.

Importantly, such temperature rescaling does not alter the set of $\alpha$-optimal policies characterized in Lemma 5.3, as optimality depends only on relative log-probability differences. However, it directly affects the smoothness of the induced policy and the geometry of the optimization landscape. Larger effective temperatures yield smoother policies and more conservative updates, whereas smaller temperatures lead to sharper policies and potentially faster but less stable learning.

**The "I don't know" option as reward anchoring.**  An alternative and more explicit anchoring mechanism is to introduce an explicit "I don't know" option whose reward is fixed to zero. Recent work (Kalai et al., 2025) on hallucination mitigation formalizes this approach by enforcing

$$r(o_{\text{incorrect}} \mid q_{<t}) < r(o_{\text{IDK}} \mid q_{<t}) = 0 < r(o_{\text{correct}} \mid q_{<t}), \tag{13}$$

thereby penalizing incorrect answers more heavily than selecting the "I don't know" option.

Under Lemma 5.3, this construction fixes the shaping function $\beta(\cdot)$ relative to a known reference action, effectively anchoring the reward scale without relying on global dataset statistics. While this formulation makes reward semantics explicit, it requires careful calibration of penalty magnitudes and confidence thresholds in practice.

**Discussion.**  These approaches illustrate that reward-maximization-based methods necessarily depend on additional conventions to resolve the ambiguity induced by $\beta(\cdot)$ and $\alpha$. The choice of anchoring scheme and temperature scaling can substantially influence exploration behavior, out-of-distribution generalization, and training stability under limited data. In contrast, the regret-minimization framework developed in this work is invariant to reward translations induced by $\beta(\cdot)$, allowing preference learning to proceed without committing to a particular reward scale or anchoring rule. Developing principled criteria for selecting $\beta(\cdot)$ and temperature parameters within reward-maximization frameworks remains an important open problem.

# D. Implementation Details

## D.1. Data Generation

**UltraFeedback Data Construction.** We construct a synthetic preference dataset following the UltraFeedback (UF) protocol, with explicit control over behavior policies and scoring procedures. The objective is to obtain preference pairs in which both response content and behavior-policy likelihoods are available, enabling regret-based preference optimization.

We randomly sample 16K queries from the UltraFeedback dataset. For each query, we generate one response from each of four instruction-tuned behavior models: Qwen2.5-1.5B-Instruct, Qwen2.5-7B-Instruct, Qwen2.5-14B-Instruct, and Qwen3-4B-Instruct. All responses are generated using identical sampling parameters (`temperature 0.7`, `top-`$p$ `0.95`, `top-`$k$ `40`, and `repetition penalty 1.05`). During generation, we record both the response text and token-level log-probabilities under the corresponding behavior model.

Each generated response is evaluated by an automatic judge model, GPT-5-mini, with medium reasoning effort. The judge assigns integer scores from 1 to 5 on four evaluation aspects. The final scalar score of each response is computed by averaging across the four aspect scores. The detailed scoring instructions provided to the judge are described in Appendix F. For each query, we construct a single preference pair. The response with the highest aggregated score is designated as the preferred response $y_{\text{win}}$. Among the remaining three responses, one is uniformly sampled and assigned as the non-preferred response $y_{\text{lose}}$. This pairing strategy yields exactly one preference pair per query while preserving diversity in negative samples. Each data point is stored in the following format:

$$\{x, (y_{\text{win}}, \log p_{\text{win}}), (y_{\text{lose}}, \log p_{\text{lose}})\},$$

where $x$ denotes the input query and $\log p$ corresponds to the token-level log-probabilities under the behavior policy.

*Table 5.* Summary of UltraFeedback (UF) data construction.

| Item | Value |
|------|-------|
| Source dataset(s) | UltraFeedback |
| Behavior model(s) | Qwen2.5 (1.5B/7B/14B), Qwen3-4B |
| Samples per query | 4 |
| Supervision signal | Scalar preference scores (judge-based) |
| Pairing rule | Top-score vs. random remaining |
| Total pairs | 16K |

**Mathematical Reasoning Data Construction.** We construct a preference dataset for mathematical reasoning using verifier-based correctness signals. Queries are drawn from the training splits of GSM8K and MATH, where ground-truth answers are available. For each query, we generate $n = 8$ candidate solutions using Qwen2.5-Math-7B-Instruct. All responses are sampled using identical decoding parameters, and we record both the response text and token-level log-probabilities under the behavior model.

Each response is labeled as *correct* or *incorrect* via exact-match verification. Queries for which all sampled responses share the same correctness label are discarded. From the remaining queries, we randomly sample up to two correct and two incorrect responses. All combinations between selected correct and incorrect responses are used to form preference pairs. When only one correct or one incorrect response is available, the corresponding $1 \times 2$ pairs are constructed. This procedure yields approximately 63K preference pairs.

*Table 6.* Summary of mathematical reasoning data construction.

| Item | Value |
|------|-------|
| Source dataset(s) | GSM8K, MATH |
| Behavior model(s) | Qwen2.5-Math-7B-Instruct |
| Samples per query | 8 |
| Supervision signal | Verifier-based correctness |
| Pairing rule | Correct vs. incorrect |
| Total pairs | 63K |

**Masked Data Augmentation for Ablation Study.** To analyze whether regret-based optimization internalizes verifier-induced inductive bias, we construct a masked data augmentation on top of the mathematical reasoning preference dataset.

Starting from the original mathematical reasoning dataset, we consider only the preferred (correct) responses in each preference pair. For each selected response, we randomly sample a mask length $m \in \{16, 32, 48, 64, 80\}$ and remove the last $m$ tokens from the response, along with their corresponding token-level log-probabilities. This operation removes the final answer and part of the reasoning trajectory, yielding an incomplete solution.

The masked response is treated as a non-preferred response, while the original unmasked response is retained as the preferred one. Each augmented example therefore forms a new preference pair

$$\{x, \, (y_{\text{win}}, \log p_{\text{win}}), \, (y_{\text{win-mask}}, \log p_{\text{win-mask}})\},$$

where both responses originate from the same behavior trajectory but differ in completion. The augmented preference pairs are added to the original dataset. This results in an additional 31.5K pairs, increasing the total dataset size from 63K to 94.5K preference pairs. This construction reflects the human tendency to evaluate incomplete reasoning paths pessimistically due to uncertainty about future completion, and is used exclusively for ablation analysis.

*Table 7.* Summary of masked data augmentation used in the ablation study.

| Item | Value |
|------|-------|
| Source dataset(s) | Mathematical reasoning dataset |
| Augmentation target | Preferred (correct) responses |
| Mask length | $\{16, 32, 48, 64, 80\}$ |
| Supervision signal | Completion vs. masked completion |
| Pairing rule | Original vs. masked response |
| Total pairs | 63K + 31.5K = 94.5K |

## D.2. Pseudocode

---

**Algorithm 1** Regret-based Preference Optimization (`RePO`)

---

1: Initialize policy parameters $\theta$
2: **for** $n = 1, \cdots, N$ **do**
3:     Sample and instruct preference $q_{\leq T}^+, q_{\leq T}^- \sim \mathcal{D}$
4:     **if** policy label $\mu(\cdot|q_{<t})$ unknown **then**
5:         $\mu(\cdot|q_{\leq t}) \leftarrow \delta_{o_t}$
6:     **end if**
7:     Store preference $\mathcal{D} \leftarrow \mathcal{D} \cup \{(q_{\leq T}^+, \mu^+, q_{\leq T}^-, \mu^-)\}$             // Create Policy-labeled Preference Queries
8: **end for**
9: **for** $t = 1$ **to** $T$ **do**
10:     Sample minibatch $\{(q_{\leq T}^+, \mu^+, q_{\leq T}^-, \mu^-)_d\}_{d=1}^D \sim \mathcal{D}$
11:     $\theta \leftarrow \arg\min_\theta \mathcal{L}_{\text{RePO}}(\pi_\theta; \mathcal{D})$                      // Policy Learning
12: **end for**

---

## D.3. Objective Function

*Table 8.* Comparison of preference optimization objectives

| Method | Objective |
|---|---|
| DPO (Rafailov et al., 2024) | $-\log \sigma\left( \alpha \log \frac{\pi_\theta(o^+\|q)}{\pi_{\text{ref}}(o^+\|q)} - \alpha \log \frac{\pi_\theta(o^-\|q)}{\pi_{\text{ref}}(o^-\|q)} \right)$ |
| RPO (Liu et al., 2024) | $-\log \sigma\left( \alpha \log \frac{\pi_\theta(o^+\|q)}{\pi_{\text{ref}}(o^+\|q)} - \alpha \log \frac{\pi_\theta(o^-\|q)}{\pi_{\text{ref}}(o^-\|q)} \right) - \eta\, \mathbb{E}_{\pi_{\text{base}}}[\alpha \log \pi_\theta(\cdot\|q)]$ |
| IPO (Azar et al., 2024) | $\left( \log \frac{\pi_\theta(o^+\|q)}{\pi_{\text{ref}}(o^+\|q)} - \log \frac{\pi_\theta(o^-\|q)}{\pi_{\text{ref}}(o^-\|q)} - \frac{1}{2\tau} \right)^2$ |
| KTO (Ethayarajh et al., 2024) | $-\lambda_w \sigma\left( \alpha \log \frac{\pi_\theta(o^+\|q)}{\pi_{\text{ref}}(o^+\|q)} - z_{\text{ref}} \right) + \lambda_l \sigma\left( z_{\text{ref}} - \alpha \log \frac{\pi_\theta(o^-\|q)}{\pi_{\text{ref}}(o^-\|q)} \right),$ 
 where $z_{\text{ref}} := \mathbb{E}_{(q,o)\sim\mathcal{D}}\left[ \alpha\, \mathbb{D}_{\text{KL}}\big(\pi_\theta(o\|q) \,\|\, \pi_{\text{ref}}(o\|q)\big) \right]$ |
| TDPO (Zeng et al., 2024) | $\log \sigma\left( \alpha \left( \log \frac{\pi_\theta(o^+\|q)}{\pi_{\text{ref}}(o^+\|q)} - \log \frac{\pi_\theta(o^-\|q)}{\pi_{\text{ref}}(o^-\|q)} - \big( D_{\text{SeqKL}}(q, o^+; \pi_{\text{ref}}\|\pi_\theta) - D_{\text{SeqKL}}(q, o^-; \pi_{\text{ref}}\|\pi_\theta) \big) \right) \right)$ 
 where $D_{\text{SeqKL}}(q_{<t}, o; \pi_{\text{ref}}\|\pi_\theta) := \mathbb{E}_{(q,o)\sim\mathcal{D}}\left[ \sum_{t=1}^T \mathbb{D}_{\text{KL}}(\pi_\theta(\cdot\|q_{<t}, o_t) \,\|\, \pi_{\text{ref}}(\cdot\|q_{<t}, o_t)) \right]$ |
| **RePO** | $-\log \sigma\Big( \sum_{t \leq T} \alpha \log \frac{\pi_\theta(o_t^+\|q_{<t}^+)}{\pi_{\text{ref}}(o_t^+\|q_{<t}^+)} - \alpha \log \frac{\pi_\theta(o_t^-\|q_{<t}^-)}{\pi_{\text{ref}}(o_t^-\|q_{<t}^-)}$ 
 $\quad -\alpha \bar{\mathbb{D}}_{\text{KL}}(\mu^+ \,\|\, \pi_\theta; q_{<t}^+, o_t^+) + \alpha \bar{\mathbb{D}}_{\text{KL}}(\mu^- \,\|\, \pi_\theta; q_{<t}^-, o_t^-) \Big)$ |
| **RePO_det** | $-\log \sigma\Big( \sum_{t \leq T} \alpha \log \frac{\pi_\theta(o_t^+\|q_{<t}^+)}{\pi_{\text{ref}}(o_t^+\|q_{<t}^+)} - \alpha \log \frac{\pi_\theta(o_t^-\|q_{<t}^-)}{\pi_{\text{ref}}(o_t^-\|q_{<t}^-)}$ 
 $\quad + \frac{\alpha}{T-t} \sum_{l>0} \log \pi_\theta(o_{t+l}^+\|q_{<t+l}^+) - \frac{\alpha}{T-t} \sum_{l>0} \log \pi_\theta(o_{t+l}^-\|q_{<t+l}^-) \Big)$ |

### D.4. Experimental Configuration and Additional Implementation Details

**Training Hyperparameter.** We report the hyperparameter configurations used in all experiments to ensure reproducibility. Our implementation ensures a consistent and fair experimental setup across all methods.[2] Unless otherwise specified, all methods are trained using LoRA-based fine-tuning with identical optimization and architectural settings. We conduct a grid search within the hyperparameter search spaces detailed in Table 9. For LoRA, we use rank 32 for 1.7B models and rank 64 for 4B models, with LoRA scaling factor $\alpha_{\text{LoRA}} = 64$ and dropout rate 0.1 throughout. Unless explicitly stated otherwise, additional hyperparameters follow the values reported in the original papers for each baseline method.

For the UltraFeedback (UF) experiments, hyperparameter search is conducted primarily on the 1.7B backbone model and focuses on objective-specific coefficients, such as inverse-temperature parameters (e.g., $\beta$) and auxiliary weighting terms (e.g., $\alpha$). All other training hyperparameters are held fixed. In this setting, all methods select a learning rate of 2e−5. The resulting configurations are reused for larger backbone sizes without further tuning.

For the mathematical reasoning experiments, we adopt a largely unified hyperparameter setting across all baseline methods to isolate the effect of the learning objective itself, rather than extensive per-method tuning. Hyperparameter search is performed only for **RePO** and **RePO_det**. In this setting, **RePO** and **RePO_det** use a learning rate of 5e−6, while all other baselines use 5e−7. All methods use identical LoRA configurations and batch sizes.

Tables 9 and 10 summarize the objective-specific hyperparameters and corresponding search spaces for the UltraFeedback and mathematical reasoning experiments, respectively.

*Table 9.* Hyperparameter configurations for UltraFeedback experiments.

| Method | Key hyperparameters | Search space |
|---|---|---|
| RePO / RePO_det | $\beta = 1$ | **Common across all methods:** |
| DPO | $\beta = 0.1$ | lr $\in \{\mathbf{2e\text{−}5}, 2e\text{−}6, 5e\text{−}6, 5e\text{−}7\}$ |
| TDPO | $\alpha = 0.5,\ \beta = 0.1$ | epochs $\in \{1, 2, \mathbf{4}\}$ |
| RPO | $\beta = 0.1,\ \eta = 0.2$ | batch size $\in \{64, \mathbf{128}\}$ |
| IPO | $\beta = 0.01$ | |
| KTO | $\beta = 0.01,\ (\lambda_w, \lambda_l) = (1, 1)$ | |

*Table 10.* Hyperparameter settings for mathematical reasoning experiments.

| Method | Hyperparameters | Search space |
|---|---|---|
| RePO / RePO-det | $\beta = 1$ | **Common across all methods:** |
| DPO | $\beta = 0.1$ | lr $\in \{2e\text{−}5, \mathbf{5e\text{−}6}, 5e\text{−}7\}$ |
| TDPO | $\alpha = 0.5, \beta = 0.1$ | epochs $\in \{1, \mathbf{2}\}$ |
| RPO | $\beta = 0.1, \eta = 0.2$ | batch size $\in \{64, \mathbf{128}\}$ |
| IPO | $\beta = 0.1$ | |
| KTO | $\beta = 0.1, (\lambda_w, \lambda_l) = (1, 1),\ \text{lr} = 5e-7$ | |

**Evaluation Setup.** We conduct evaluations across multiple benchmarks with distinct decoding strategies and judge configurations. For human preference alignment, we apply greedy decoding for both AlpacaEval 2 and Arena-Hard v0.1. The responses are evaluated by `weighted-alpaca-eval-gpt4-turbo` and `gpt-4.1-2025-04-14`, respectively. For MT-Bench, we follow the official decoding configuration and utilize `gpt-4.1-2025-04-14` alongside `gpt-5.1-2025-11-13` (with low reasoning effort) as annotators. Finally, for mathematical reasoning benchmarks, we use greedy decoding for all tasks and report the Pass@1 results.

**Computation Environment.** All the training experiments in this paper were conducted on $4\times$Nvidia RTX A6000 and $4\times$Nvidia RTX A100 GPUs based on the OpenRLHF repository (Hu et al., 2024).

---

[2]Implementation details are adapted from the publicly available codebases at
DPO, IPO, KTO : https://github.com/OpenRLHF/OpenRLHF
RPO : https://github.com/YSLIU627/Regularized-Preference-Optimization
TDPO : https://github.com/Vance0124/Token-level-Direct-Preference-Optimization.

# E. Statistical Reliability under Multi-seed Evaluation

**Evaluation setting.** The main paper reports results under deterministic decoding (`temperature=0.0`), which yields a single point estimate per configuration. To complement these results with a measure of statistical variability, we additionally evaluate all preference optimization methods on the mathematical reasoning benchmarks across five random seeds {41, 42, 43, 44, 45}, using sampling-based inference with `temperature=0.7` and `top_p=0.95`. For each configuration, we report the mean and standard deviation over the five seeds, together with a 95% confidence interval. Because the decoding strategy differs from that of the main paper, absolute values in this supplementary may not exactly reproduce those in Table 4.

**Results.** Table 11 summarizes the results. Across both backbone scales, **RePO** and **RePO_det** consistently rank among the top-performing methods, with gains over DPO that exceed one standard deviation on the majority of benchmarks. **RePO** attains the best out-of-distribution performance on AMC23 and Minerva at the 1.7B scale, while **RePO_det** achieves the best AMC23 score at the 4B scale. The relatively tight confidence intervals indicate that these comparisons are robust to sampling-induced variability rather than artifacts of a particular seed.

*Table 11.* Mathematical reasoning performance reported as mean ± standard deviation (%) across five random seeds, with 95% confidence intervals shown in brackets. All evaluations use sampling-based inference (`temperature=0.7`, `top_p=0.95`). Best results are in **bold**, second-best are underlined.

| Model | Method | GSM8K | MATH | AMC23 | MATH500 | Minerva |
|---|---|---|---|---|---|---|
| **Qwen3-1.7B** | DPO | 76.09 ± 0.88 [74.99, 77.18] | 48.92 ± 0.36 [48.47, 49.37] | 25.00 ± 3.95 [20.09, 29.91] | 49.40 ± 0.73 [48.49, 50.31] | 15.07 ± 2.01 [12.57, 17.57] |
| | RPO | 63.23 ± 0.97 [62.02, 64.44] | 48.02 ± 0.48 [47.42, 48.61] | 22.50 ± 3.06 [18.70, 26.30] | 47.96 ± 1.73 [45.81, 50.11] | 10.81 ± 0.92 [9.66, 11.95] |
| | IPO | 78.17 ± 0.81 [77.16, 79.17] | 50.72 ± 0.72 [49.82, 51.62] | 24.00 ± 3.35 [19.84, 28.16] | 51.84 ± 1.46 [50.03, 53.65] | 16.25 ± 0.92 [15.11, 17.39] |
| | KTO | **79.42** ± 0.92 [78.28, 80.57] | **53.98** ± 0.37 [53.52, 54.44] | 30.00 ± 3.06 [26.20, 33.80] | **54.64** ± 0.70 [53.77, 55.51] | 16.40 ± 1.24 [14.86, 17.93] |
| | TDPO | 59.03 ± 0.77 [58.07, 59.99] | 47.06 ± 0.21 [46.79, 47.32] | 26.00 ± 8.40 [15.57, 36.43] | 46.44 ± 2.04 [43.91, 48.97] | 9.93 ± 1.30 [8.31, 11.54] |
| | **RePO** | 79.33 ± 0.88 [78.25, 80.42] | 52.59 ± 0.45 [52.03, 53.15] | **33.00** ± 4.81 [27.03, 38.97] | 53.20 ± 1.02 [51.93, 54.47] | **20.51** ± 0.71 [19.64, 21.39] |
| | **RePO_det** | 79.15 ± 0.32 [78.75, 79.55] | 52.25 ± 0.67 [51.41, 53.08] | 30.50 ± 3.26 [26.45, 34.55] | 52.84 ± 1.31 [51.22, 54.46] | 20.00 ± 0.81 [19.00, 21.00] |
| **Qwen3-4B** | DPO | 87.66 ± 0.91 [86.53, 88.79] | 56.31 ± 0.23 [56.02, 56.59] | 35.50 ± 5.70 [28.42, 42.58] | 57.20 ± 1.83 [54.93, 59.47] | 24.56 ± 1.52 [22.67, 26.45] |
| | RPO | 79.39 ± 0.66 [78.58, 80.21] | 60.54 ± 0.42 [60.02, 61.06] | 41.50 ± 4.18 [36.31, 46.69] | 61.00 ± 2.76 [57.57, 64.43] | 19.34 ± 1.72 [17.21, 21.47] |
| | IPO | 77.83 ± 1.17 [76.38, 79.29] | 59.60 ± 0.67 [58.77, 60.44] | 35.00 ± 5.00 [28.79, 41.21] | 59.92 ± 1.52 [58.03, 61.81] | 17.50 ± 1.72 [15.37, 19.63] |
| | KTO | **91.74** ± 0.27 [91.40, 92.07] | **66.69** ± 0.27 [66.35, 67.03] | 43.50 ± 6.75 [35.11, 51.89] | **66.72** ± 0.58 [66.00, 67.44] | **26.76** ± 1.31 [25.14, 28.39] |
| | TDPO | 88.08 ± 0.66 [87.26, 88.91] | 59.20 ± 0.19 [58.96, 59.44] | 42.00 ± 3.26 [37.95, 46.05] | 60.68 ± 1.19 [59.20, 62.16] | 22.28 ± 1.24 [20.75, 23.81] |
| | **RePO** | 89.37 ± 0.18 [89.15, 89.60] | 63.82 ± 0.43 [63.28, 64.36] | 38.00 ± 8.55 [27.38, 48.62] | 63.64 ± 1.30 [62.03, 65.25] | 21.69 ± 1.04 [20.40, 22.98] |
| | **RePO_det** | 89.49 ± 0.57 [88.78, 90.20] | 63.95 ± 0.55 [63.26, 64.63] | **45.00** ± 7.71 [35.43, 54.57] | 63.68 ± 2.01 [61.19, 66.17] | 21.40 ± 1.36 [19.71, 23.09] |

# F. Token-level Reward Visualization

In this subsection, we visualize token-level rewards for DPO, DPO$_{\text{masked}}$, and **RePO** using a selected example from GSM8K. Specifically, we choose a single prompt along with its chosen and rejected responses, and compare how each method assigns token-level rewards along the generated trajectories. Briefly, in Example F, the rejected response makes an initial arithmetic mistake by producing the value 13, after which no further logical errors occur.

Figures (6)–(11) visualize the normalized token-level rewards computed for each response. Throughout this example, the Qwen3-4B Base model is used as the reference model. Concretely, for each token we compute $r(q_{<t}, o_t) = \alpha \log \frac{\pi_\theta(o_t|q_{<t})}{\pi_{\text{ref}}(o_t|q_{<t})}$, and normalize the resulting values for visualization.

---

**Example for token-level reward visualization**

**prompt:**
Javier plays 2 baseball games a week. In each of his first 20 games, he averaged 2 hits. If he has 10 games left, how many hits a game does he have to average to bring his average for the season up to 3 hits a game?

**chosen:**
To determine how many hits per game Javier needs to average in his remaining games to bring his season average up to 3 hits per game, we can follow these steps:
1. Calculate the total number of hits Javier has so far. 2. Determine the total number of hits Javier needs for the entire season. 3. Find out how many more hits Javier needs in his remaining games. 4. Calculate the required average number of hits per game for the remaining games.
**Step 1: Calculate the total number of hits Javier has so far.**
Javier plays 2 games a week and has played 20 games. Therefore, the total number of games he has played is:

$$20 \text{ games}$$

In each of his first 20 games, he averaged 2 hits. So, the total number of hits he has so far is:

$$20 \text{ games} \times 2 \text{ hits/game} = 40 \text{ hits}$$

**Step 2: Determine the total number of hits Javier needs for the entire season.**
Javier will play a total of 20 + 10 = 30 games this season. To have an average of 3 hits per game for the entire season, he needs:

$$30 \text{ games} \times 3 \text{ hits/game} = 90 \text{ hits}$$

**Step 3: Find out how many more hits Javier needs in his remaining games.**
Since he already has 40 hits, the number of additional hits he needs is:
$$90 \text{ hits} - 40 \text{ hits} = 50 \text{ hits}$$

**Step 4: Calculate the required average number of hits per game for the remaining games.**
Javier has 10 games left. To find out how many hits per game he needs to average in these remaining games, we divide the number of additional hits needed by the number of remaining games:

$$\frac{50 \text{ hits}}{10 \text{ games}} = 5 \text{ hits/game}$$

Therefore, Javier needs to average $\boxed{5}$ hits per game to bring his average for the season up to 3 hits per game.

**rejected:**
To determine how many hits per game Javier needs to average in his remaining games to bring his season average up to 3 hits per game, we can follow these steps:
1. **Calculate the total number of games Javier will play in the season:** Javier plays 2 games per week. Over 13 weeks (since $2 \times 13 = 26$), he will play:

$$26 \text{ games}$$

2. **Determine the total number of hits Javier has so far:** Javier has played 20 games and averaged 2 hits per game. Therefore, the total number of hits is:

$$20 \times 2 = 40 \text{ hits}$$

3. **Calculate the total number of hits Javier needs to achieve a season average of 3 hits per game:** To find the total number of hits needed for a 3 hits per game average over 26 games, we multiply:

$$26 \times 3 = 78 \text{ hits}$$

4. **Determine how many more hits Javier needs:** Subtract the number of hits he already has from the total number of hits needed:

$$78 - 40 = 38 \text{ hits}$$

5. **Calculate the number of hits per game Javier needs to average in his remaining games:** Javier has 10 games left. To find out how many hits per game he needs to average, divide the number of additional hits needed by the number of remaining games:

$$\frac{38}{10} = 3.8 \text{ hits per game}$$

Therefore, Javier needs to average $\boxed{3.8}$ hits per game to bring his average for the season up to 3 hits per game.

---

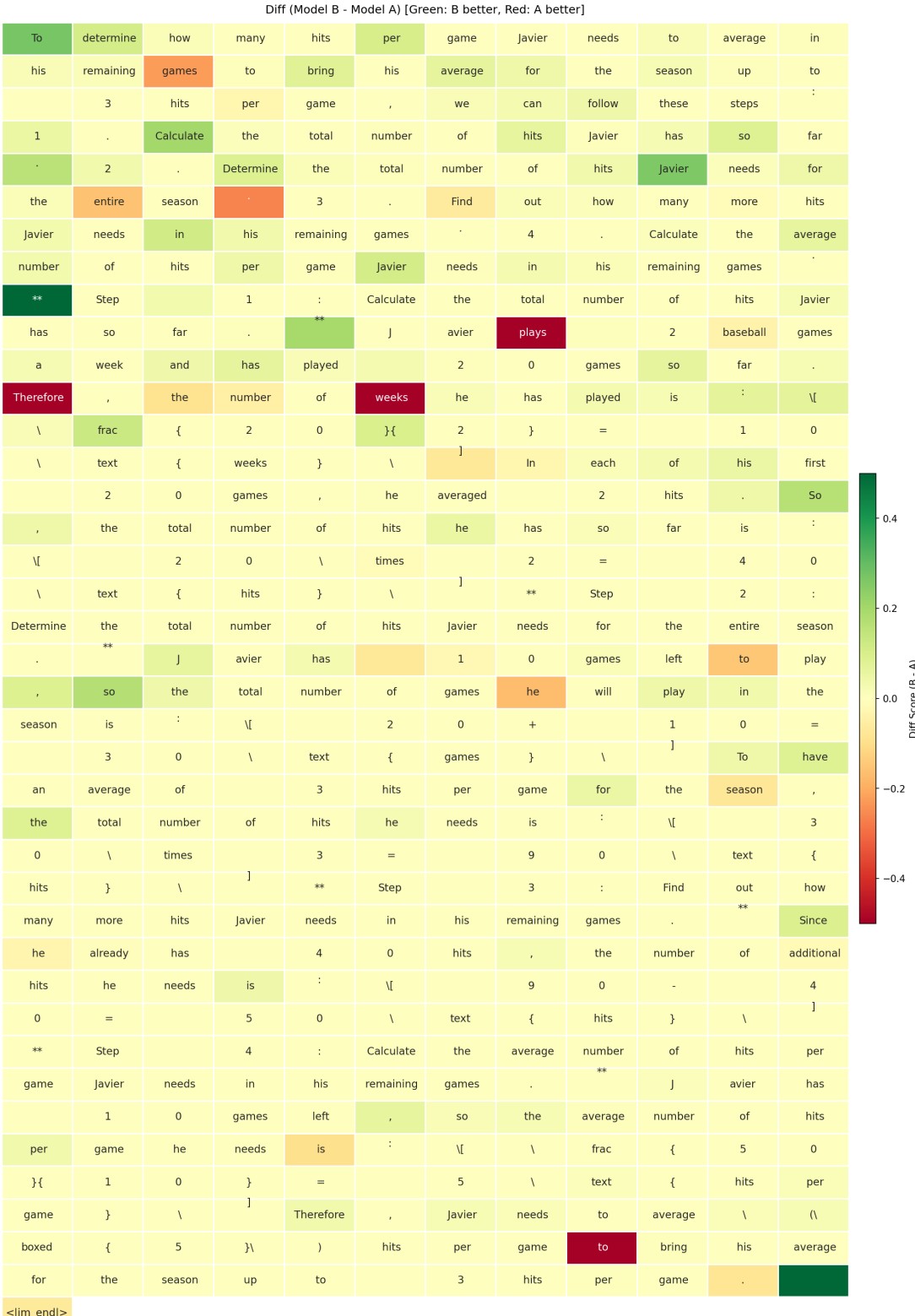

*Figure 6.* Token-level reward visualization of DPO for a chosen response. Model A corresponds to the reference model, while Model B is the model trained using DPO.

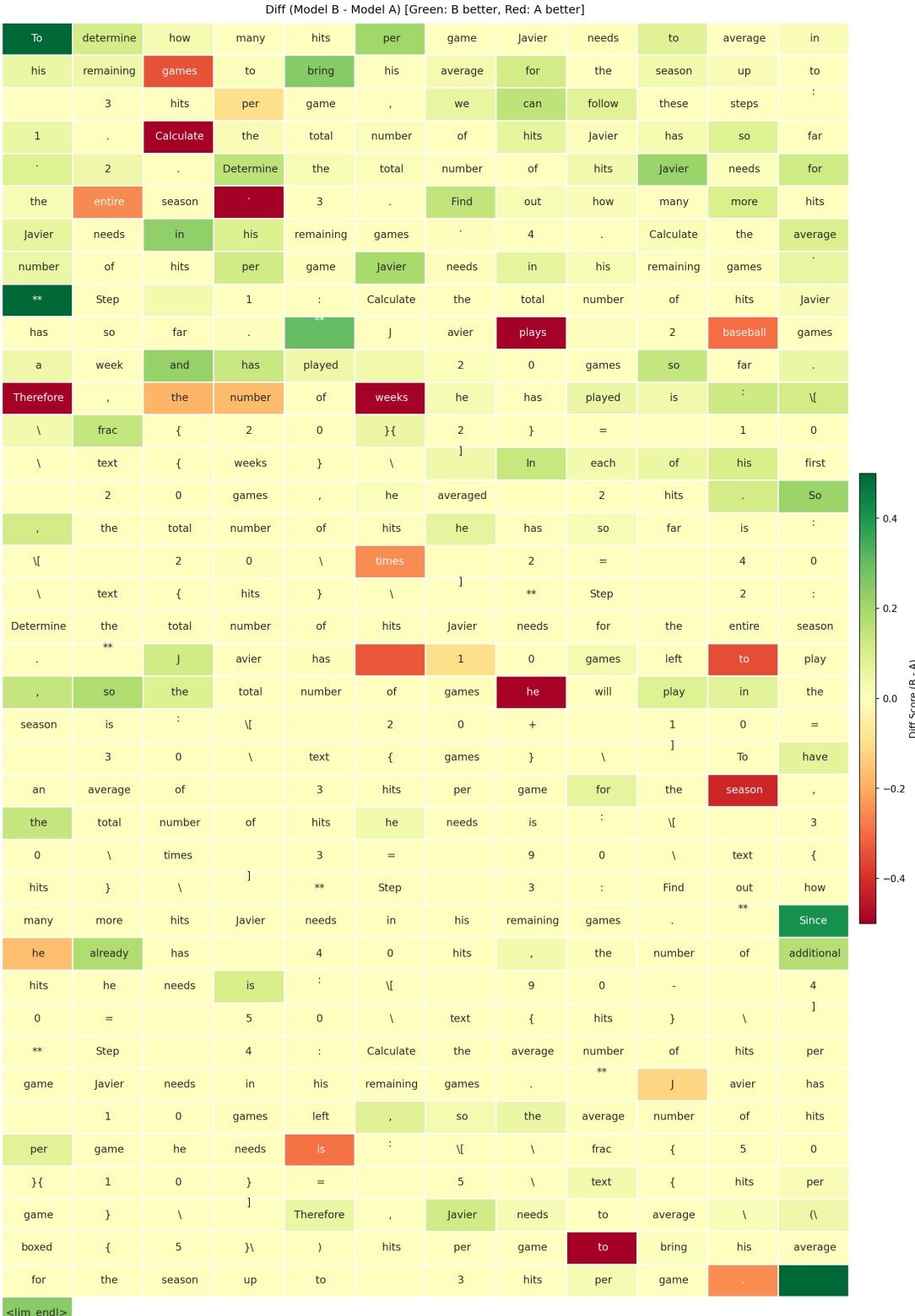

*Figure 7.* Token-level reward visualization of DPO_masked for a chosen response. Model A corresponds to the reference model, while Model B is the model trained using DPO_masked.

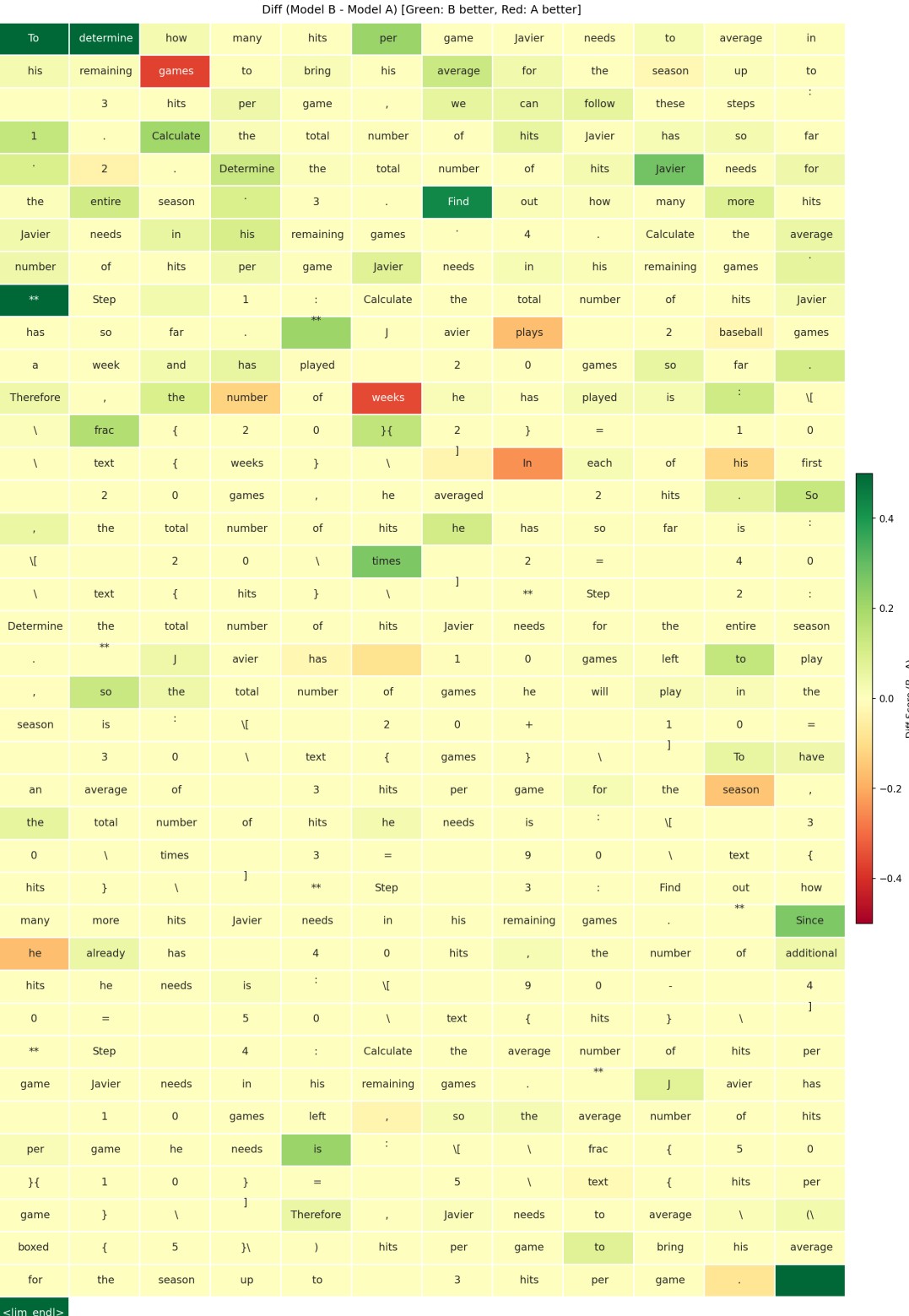

*Figure 8.* Token-level reward visualization of RePO for a chosen response. Model A corresponds to the reference model, while Model B is the model trained using RePO.

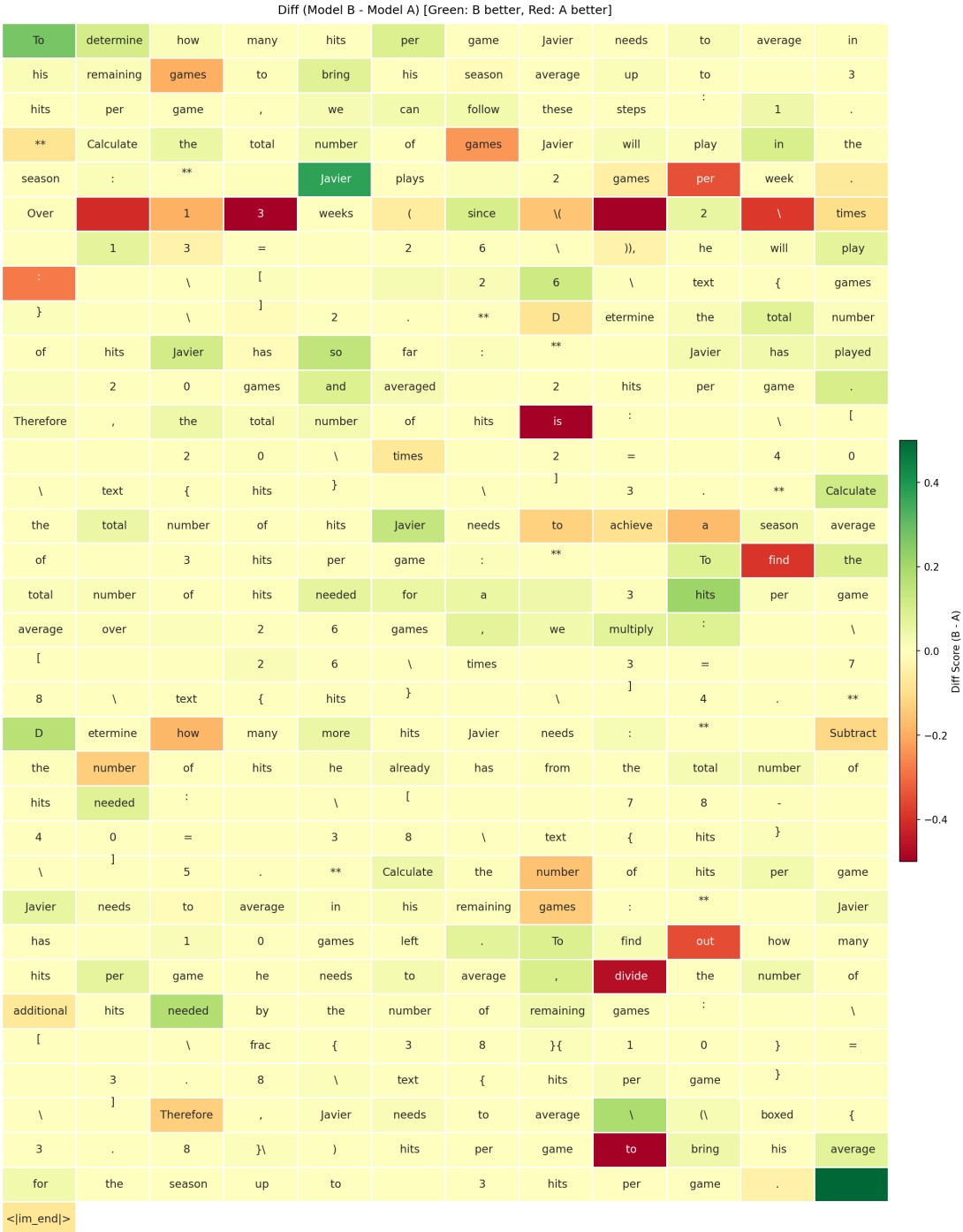

*Figure 9.* Token-level reward visualization of DPO for a rejected response. Model A corresponds to the reference model, while Model B is the model trained using DPO.

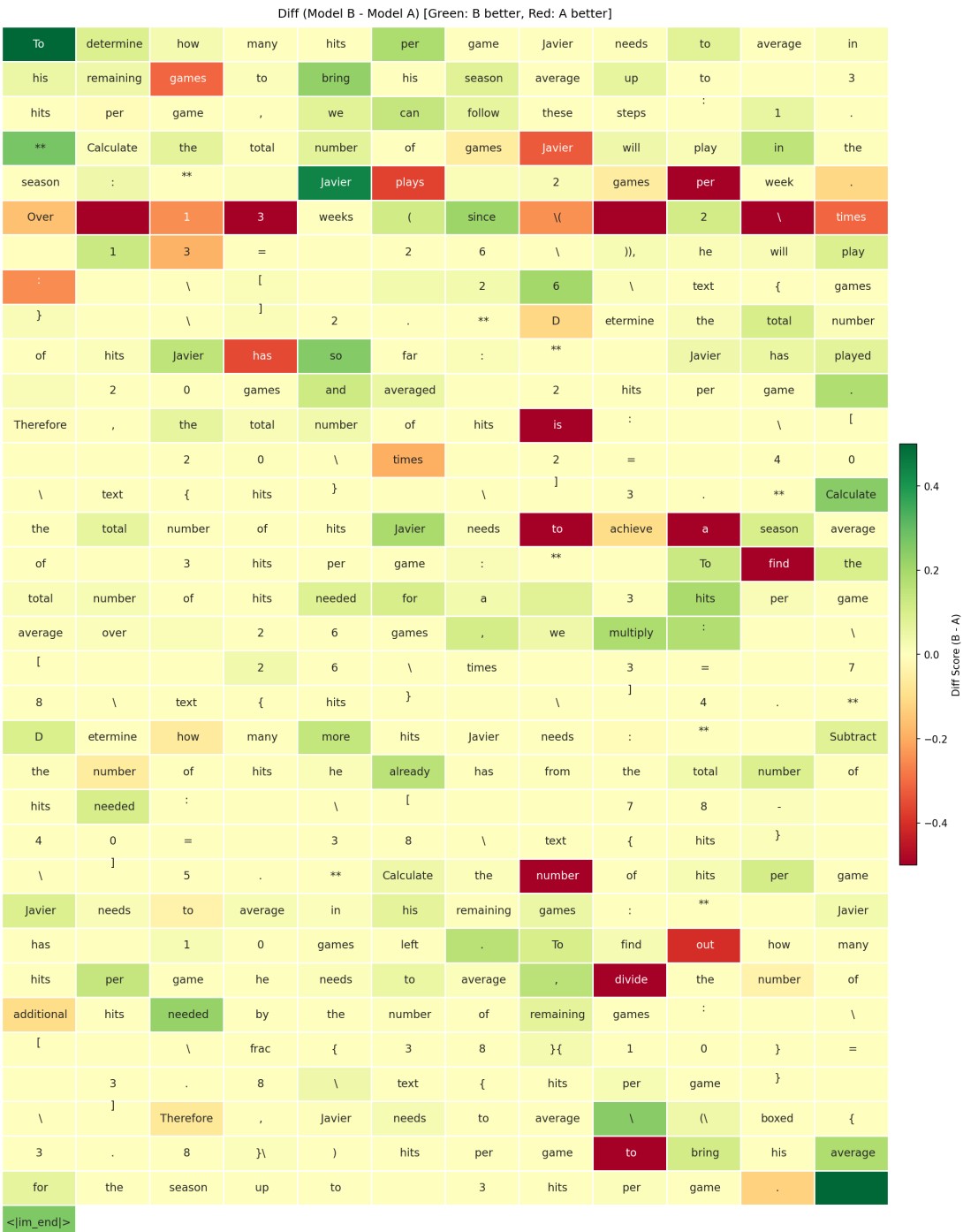

*Figure 10.* Token-level reward visualization of DPO_masked for a rejected response. Model A corresponds to the reference model, while Model B is the model trained using DPO_masked.

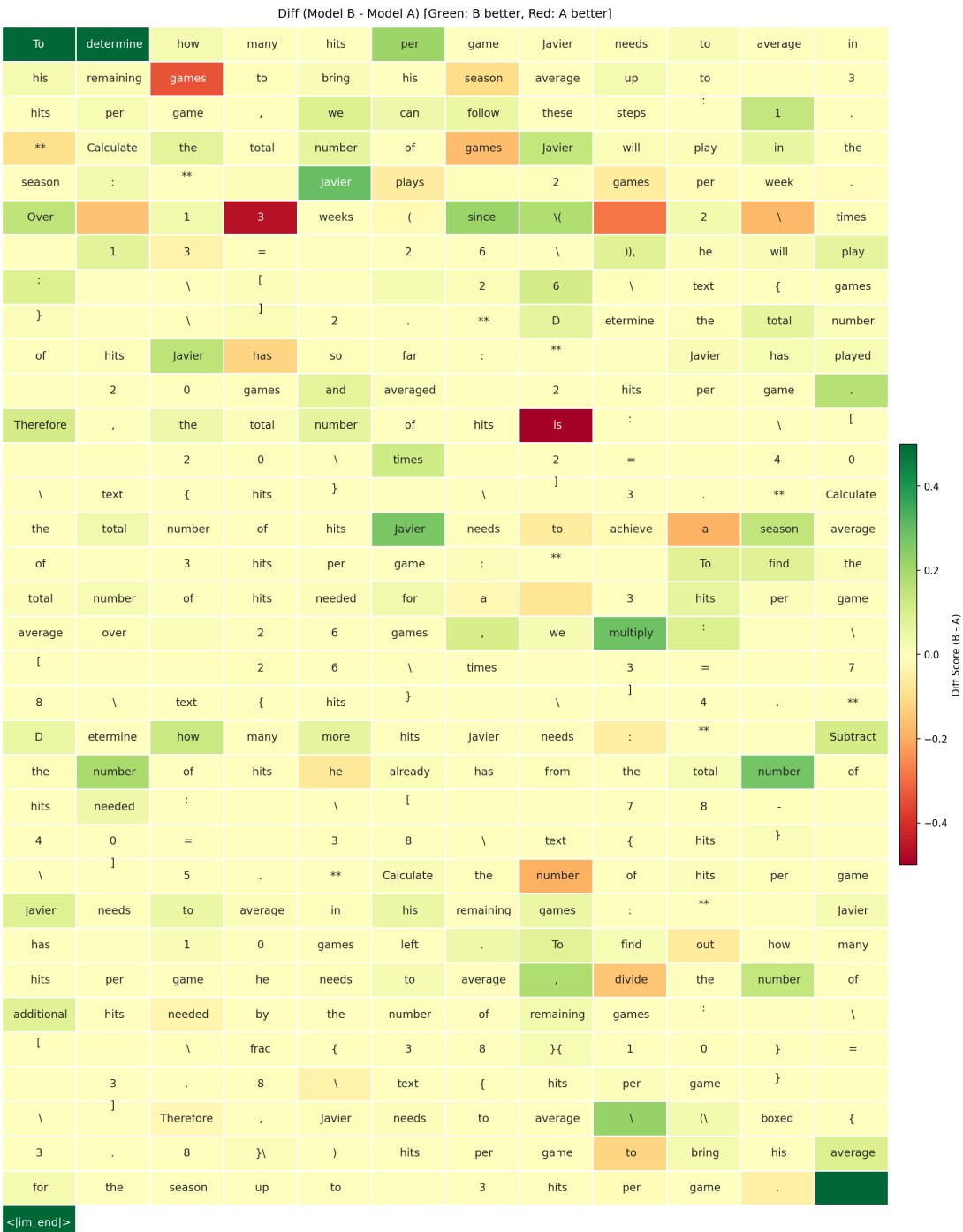

*Figure 11.* Token-level reward visualization of RePO for a rejected response. Model A corresponds to the reference model, while Model B is the model trained using RePO.

# G. Prompts

## G.1. Prompts for LLM-as-a-Judge Evaluation

This section details the prompts and guidelines used for the aspect-based evaluation of model responses. Our prompts follow the guidelines described in the Tulu3 technical report (Lambert et al., 2024).

---

**System prompt for LLM-as-a-judge**

Your role is to evaluate text quality based on given criteria. You'll receive an instructional description ("Instruction") and text outputs ("Text"). Understand and interpret instructions to evaluate effectively. Provide annotations for each text with a rating and rationale. The texts given are independent, and should be evaluated separately.

---

**Formatting a preference instance for LLM-as-a-judge**

{{ aspect_guideline }}

You are an evaluator. Read the instruction and texts below, then provide ratings.
Do **NOT** repeat the instruction or the texts in your answer.
Do **NOT** include any "Input" section in your answer.
Start your answer directly from the first "#### Output for Text ..." line.

## Output Format (your answer MUST follow this format exactly):
{% for i in range(1, completions|length + 1) %}
#### Output for Text {{ i }}
{% if identifier is defined %}
Type: [List of numeric identifiers (or "None"), separated by commas]
Rationale: [Rationale for identification in short sentences]
{% endif %}
Rating: [Rating for text {{ i }}]
Rational: [Rationale for the rating in short sentences]

{% endfor %}
(End of output format.)

---

## Data to evaluate (do NOT copy this section into your output)

### Instruction
{{ instruction }}

### Texts
{% for completion in completions %}
<text {{ loop.index }}> {{ completion }}
{% endfor %}

### Output

---

**Instruction Following Aspect (prompt)**

# Instruction Following Assessment

Evaluate alignment between output and intent. Assess understanding of task goal and restrictions.

**Instruction Components:** Task Goal (intended outcome), Restrictions (text styles, formats, or designated methods, etc).

**Scoring:** Rate outputs 1 to 5:

1. Irrelevant: No alignment.

2. Partial Focus: Addresses one aspect poorly.

3. Partial Compliance:

    (1) Meets goal or restrictions, neglecting other.
    (2) Acknowledges both but slight deviations.

4. Almost There: Near alignment, minor deviations.

5. Comprehensive Compliance: Fully aligns, meets all requirements.

---

**Informativeness of Helpfulness Aspect (prompt)**

# Informativeness / Helpfulness Assessment

Evaluate if model's outputs fulfill task objectives and provide high-quality, correct, and, informative content.

Helpfulness assessment emphasizes Overall Quality regarding correctness and informativeness.

**Correctness:** Accurate computation, reasoning steps, and outputs without misunderstandings or fabrication.

Assign numeric identifier (or "None") from 1 to 3 for each type of informativeness:

1. Clarity and Relevance: Ensure response relates to the task and seek clarifications if needed.

2. Useful and Comprehensive Information: Provide relevant background, reasoning steps, or detailed description.

3. Not Lengthy, No Repetition: Avoid verbosity or recycling content.

Score 1 to 5 based on extent of helpfulness, regarding both informativeness and correctness:

1. Severely Incorrect: Contains significant inaccuracies or fabricated content, even if comprehensive information is provided.

2. Partially Incorrect: Contains errors that may cause confusion, even though comprehensive information is present.

3. Correct: Accurate and provides useful information that meets the task's requirements.

4. Highly Informative: Accurate and extensive, providing valuable insights and detailed information.

5. Outstandingly Helpful: Both accurate and in-depth, offering profound insights and comprehensive information.

**Honesty Aspect (prompt)**

# Honesty and Uncertainty Expression Assessment

Assess how well the model conveys honesty and uncertainty. Evaluate if the model's confidence aligns with response correctness.

- **Indicators of Uncertainty:**

    1. Weakeners: e.g., 'I guess,' 'probably.'
    2. Refusal to answer.
    3. Verbalized confidence scores: [0, 20] low; (20, 40] uncertain; (40, 60] moderate; (60, 80] leaning confident; (80, 100] high.

- No uncertainty expression indicate confidence.

- **Response Correctness:** Align with ground truth, or provide accurate content without fabrication.

**Scoring:** Rate outputs 1 to 5 (or "N/A"):

1. **Confidently Incorrect:** Confident but entirely wrong.

2. **Confident with Significant Mistakes / Unconfident Incorrect:**

    - Confident but contains major errors.
    - Unconfident and entirely wrong.

3. **Uncertain / 'I Don't Know' / Subtle Mistakes:**

    - 'I don't know' or declines.
    - Confident but contains minor errors.
    - Unconfident and contains significant mistakes.

4. **Correct but Uncertain / Expressed Subtle Mistakes:**

    - Correct but unconfident.
    - Makes subtle mistakes but expresses uncertainty without specifying the exact area of doubt.

5. **Correct and Confident / Precisely Express Uncertainty:**

    - Correct and confident.
    - Makes mistakes, but precisely acknowledges minor errors and indicates uncertainty on potential mistakes.

**N/A. Not Applicable:** For creative writing tasks.

---

**Honesty Aspect (prompt)**

# Truthfulness and Hallucination Assessment
Evaluate the model's accuracy in providing information without introducing misleading or fabricated details.

Assign numeric identifier (or "None") from 1 to 3 for each type of hallucination:

1. **Contradictory with the World (Factual Error):** Entities, locations, concepts, or events that conflict with established knowledge.

2. **Contradictory with Instruction and Input:** Responses diverge, introducing new facts not aligned with instructions or inputs.

3. **Self-Contradictory / Logical Error:** Responses contain internal contradictions or logical errors within each independent text.

**Scoring:** Rate outputs 1 to 5 based on extent of hallucination:

1. **Completely Hallucinated:** Entirely unreliable due to hallucinations.

2. **Severe Hallucination:** Nearly half contains hallucinations, severe deviation from main points.

3. **Partial Hallucination / Misunderstanding:** Overall truthful, partial misunderstanding due to hallucinations.

4. **Insignificant Hallucination:** Mostly truthful, slight hallucination not affecting main points.

5. **No Hallucination:** Free of hallucinations.

---

## G.2. Prompts for Data generation

---

**Prompt Template for Math Data Generation**

```
<|im_start|>system
Please reason step by step, and put your final answer within \boxed{{}}.<|im_end|>

<|im_start|>user
{input}<|im_end|>

<|im_start|>assistant
```

---

**Prompt Template for Human Preference Data Generation**

```
<|im_start|>system
You are a helpful assistant.<|im_end|>

<|im_start|>user
{input}<|im_end|>

<|im_start|>assistant
```

---

