# OpenReview forum: "A Regret Minimization Framework on Preference Learning  in Large Language Models"
_ICML.cc/2026/Conference — ICML 2026 spotlight_

### Official Review · Reviewer_vD71 · 2026-02-17

**Soundness:** 4
**Presentation:** 3
**Significance:** 3
**Originality:** 3
**Overall Recommendation:** 5
**Confidence:** 3

**Summary:**

This paper proposes an alternative interpretation of human feedback in the context of RLHF, and formalizes the optimization problem in terms of \emph{regret minimization} rather than immediate reward maximization. Within the KL regularization framework, the regret-based preference optimization (RePO) admits a closed-form update rule. Building on these theoretical results, a series of experiments is conducted on math reasoning and human preferences alignment datasets, to explore three questions: (i) can RePO achieve high performance (compared to other methods) on both benchmarks? (ii) How effective is RePO in the absence of behavior-policy information? and (iii) does RePO outperform the reward maximization paradigm in terms of sample efficiency?

**Compliance With Llm Reviewing Policy:**

Affirmed.

**Final Justification:**

**EDIT:** I have raised my score after reading the rebuttal (4->5). I have raised several concerns regarding the significance and scope of the results, all of which have been adequately addressed by the authors' during the rebuttal.

**Key Questions For Authors:**

See 'weaknesses' above.

**Limitations:**

yes

**Strengths And Weaknesses:**

## Strengths
- The core idea of the paper (reframing preference optimization in terms of regret rather than reward) is novel (at least as far as I know) and well motivated and justified. In particular, I think that the authors do a great job in illustrating the limitations of reward maximization in Sec. 4.1.
- The fact that the proposed approach admits a closed-form update rule in the KL regularization paradigm makes it applicable, and the theoretical derivation seems sound as far as I can tell.

## Weaknesses
- The empirical results are presented without confidence intervals/stdv, so it is hard to tell whether differences in performance are statistically significant.
- The fact that all experiments are conducted with Qwen models limits their generalizability. While it is understood that conducting further experiments is costly, I think that even some preliminary results with other models (e.g., Llama/Gemma) would strengthen the paper.
- While the quantitative results are comprehensive, the paper could benefit from supporting them with some output trajectory examples that capture and demonstrate the difference between models trained with regret minimization compared to reward maximization, or, even better, conduct some error analysis and try to identify common patterns that separate the two approaches.

---

> ### Author Rebuttal · Authors · 2026-03-30
>
> We sincerely thank the reviewer for the thoughtful and constructive feedback. We address each question below.
>
> ## Q1. Statistical significance of results
> For the results reported in the main paper, we used *deterministic inference* with temperature set to 0. Accordingly, confidence intervals and standard deviations were not applicable in that setting.
>
> To address this concern more directly, we additionally conducted 5-seed experiments on the math-reasoning benchmark with temperature set to 0.7. Due to the evaluation API cost of human preference benchmarks, we prioritized the math domain for this additional analysis. The results remain consistent: RePO continues to show stable improvements.
>
>
> |Model|Algorithm|GSM8K|MATH|AMC23|MATH500|Minerva|
> |----------|-----------:|-------------:|-------------:|-------------:|-------------:|-------------:|
> |Qwen3-1.7B|DPO|76.09±0.88|48.92±0.36|25.00±3.95|49.40±0.73|15.07±2.01|
> |Qwen3-1.7B|**RePO**|**79.33±0.88**|**52.59±0.45**|**33.00±4.81**|**53.20±1.02**|**20.51±0.71**|
> |Qwen3-1.7B|**RePO_det**|79.15±0.32|52.25±0.67|30.50±3.26|52.84±1.31|20.00±0.81|
>
> |Model|Algorithm|GSM8K|MATH|AMC23|MATH500|Minerva|
> |--------|-----------:|-------------:|-------------:|-------------:|-------------:|-------------:|
> |Qwen3-4B|DPO|87.66±0.91|56.31±0.23|35.50±5.70|57.20±1.83|**24.56±1.52**|
> |Qwen3-4B|**RePO**|89.37±0.18|63.82±0.43|38.00±8.55|63.64±1.30|21.69±1.04|
> |Qwen3-4B|**RePO_det**|**89.49±0.57**|**63.95±0.55**|**45.00±7.71**|**63.68±2.01**|21.40±1.36|
>
>
> ## Q2. Generalization across model families
> Regarding generalization beyond Qwen, Section 7.2 of the main paper already includes results on Llama-family models, showing that RePO is not tied to a specific model family. Extending the experiments in that subsection, we further conducted additional experiments on Gemma and Llama models of different sizes, and again observed consistent gains.
>
>
> |Backbone|Method|GSM8K|MATH|MATH500|Minerva|AMC23|
> |-------------|-------:|-------------:|-------------:|-------------:|------------:|------------:|
> |Gemma-2-2b-it|DPO|45.56±0.33|20.92±0.64|21.07±1.10|3.68±1.95|4.17±1.44|
> |Gemma-2-2b-it|RePO_det|**58.23±0.91**|**25.45±0.31**|**24.00±1.40**|**8.70±2.39**|**5.00±5.00**|
>
> |Backbone|Method|GSM8K|MATH|MATH500|Minerva|AMC23||
> |--------------------|-------:|-------------:|-------------:|-------------:|------------:|-------------:|-|
> |Llama3.2-1B-Instruct|DPO|35.56±0.65|19.58±0.17|18.87±1.21|**5.39±2.02**|7.50±2.50||
> |Llama3.2-1B-Instruct|RePO_det|**36.75±0.98**|**20.45±0.26**|**19.33±0.95**|5.15±0.97|**11.67±2.89**|
>
>
> ## Q3. Qualitative differences and interpretability
> For the qualitative difference between regret minimization and reward maximization, the clearest evidence appears at the token-level rather than from output-only comparisons. For this reason, we refer the reviewer to the comparison between Figure 7 and Figure 9 in Appendix Section E.
>
> These figures analyze a rejected math trajectory in which an early mistake leads to an incorrect final answer, even though much of the later response is locally correct. This example illustrates the distinction clearly:
> - DPO penalizes the rejected trajectory broadly and uniformly across timesteps,
> - whereas RePO applies a more selective, fine-grained penalty.
> This difference is directly visible in the log-probability ratio patterns shown in the figures.
>
>
> To further support this, we also performed an additional error-analysis-style study based on the masked-augmentation setup in Section 7.3. We assume that DPO would require more from additional fine-grained supervision, whereas RePO already incorporates much of this signal by its structural design (Lemma 6.1). The empirical results support that, while RePO remains relatively stable, DPO requires additional amounts of augmented data to achieve this inductive bias.
>
>
> | Augmented tokens | 0 | 5k | 10k | 15k | 20k | 25k | 30k |
> |--- |---:|---:|---:|---:|---:|---:|---:|
> | DPO-1.7B  | 72.86 | 73.62 | 74.22 | 76.04 | 77.48 | 78.47 | 78.62 |
> | RePO-1.7B | 79.30 | 79.76 | 79.53 | 80.14 | 79.76 | 80.59 | 79.53 |
> | DPO-4B  | 86.50 | 87.34 | 88.10 | 89.01 | 89.76 | 88.63 | 89.69 |
> | RePO-4B | 90.60 | 91.28 | 90.83 | 90.45 | 91.36 | 90.75 | 91.05 |
>
>
>  > For visualizations and more detailed results, please refer to [ [this link](https://github.com/Anonymous-account-for-openreview/supplementary-material-for-icml2026/blob/main/README.md) ].
>
>
> We thank the reviewer once again for the careful and detailed comments. We will incorporate additional discussions on statistical significance, cross-model generalization, and qualitative analysis in the revision to further strengthen the paper.

---

> > ### Author Rebuttal · Reviewer_vD71 · 2026-04-02
> >
> > All of my concerns have been addressed, and I believe that the paper should be accepted. Therefore, I am raising my score (4->5).

---

> > > ### Author Response · Authors · 2026-04-02
> > >
> > > Thank you very much for your thoughtful follow-up and positive reassessment.
> > >
> > > We sincerely appreciate your helpful feedback. Your comment on strengthening the empirical results helped us improve the paper, and we will incorporate these additions in the revised version.

---

### Official Review · Reviewer_jKZb · 2026-02-18

**Soundness:** 3
**Presentation:** 2
**Significance:** 2
**Originality:** 2
**Overall Recommendation:** 3
**Confidence:** 4

**Summary:**

The authors suggest to use an alternative to the widely used Bradley-Terry model replacing the reward of the trajectory with the regret along the trajectory which formalizes the notion of "how much better the future trajectory could have been if the best policy was followed".

With this change they derive an algorithm RePO which performs very convincigly in prcatice.

**Compliance With Llm Reviewing Policy:**

Affirmed.

**Final Justification:**

The authors addressed my concern about the step-by-step supervision.

However, several weaknesses remain in my opinion:
1)  I think that using the word regret for an offline problem is confusing. Moreover, I think that the notion the authors decided to use is more similar to simple regret rather than the standard regret. Indeed, their sum is over temporal steps within one episode, not across episodes as in standard online learning.  Therefore, we can view the problem as making a unique decision, which is a stationary policy.  Unfortunately, the authors decided not to follow my suggestion of presenting things in terms of simple regret.
2) RePO_det sounds more like a heuristic rather than a principled approach to unknown behavioural policies.
3) I think that the theoretical analysis is incomplete. There is no provable advantage of using REPO over DPO or more standard preference-based methods. The authors also disagreed on presenting this section as Algorithm derivations. I have inspected their proof, and I do not think that they are new compared to standard facts in entropy regularized MDPs/contextual bandits. Therefore, I am still convinced that the content of this section is not very valuable.

For the reasons above, I maintain the weak reject score that I assigned initially.

**Key Questions For Authors:**

1) Is it reasonable to assume that the preference oracle can provide preference bits along the observed trajectories at each time step of the chain of thought ? I think that it is usually more common that the preference bit is given to the learner, looking at the whole trajectory.

2) When one can access the preference oracle once per trajectory (i.e. the problem boils down to a bandit setting), then the regret minimization framework and the reward maximization coincide. Therefore, in most practical situations, REPO would boil down to DPO, right ? The methods differ only when a way more costly preference oracle (which can be queried along the observed trajectory) is available. Is this correct?

3) Can you point your definition of regret to any reference in the online learning literature ?
It vaguely resembles to me the regret definition in average reward definition in adversarial MDPs but the authors should provide a clear pointer to the literature. The reference Lattimore \& Szepesvari defines regret differently, with a sum over episodes of learning, an aspect which is missing in this submission.

**Limitations:**

Please see the questions.

My main reason for a borderline reject score is that the preference bits need to be collected at different temporal stages of the trajectories. A situation rarely encountered in practice, when the preference bit is given once, comparing a pair of full trajectories.
If the authors clarify this point or convince me that this preference oracle is still often available in practice, I will increase my score.

As a last thing, I would rename the section Theoretical results as algorithm derivation. In fact, none of your theorem shows a theoretical property of the algorithm or a provable advantage over DPO. These are just formulas that you need to derive the REPO loss function. Please correct me if I am wrong.

**Strengths And Weaknesses:**

The experiments are vast and convincing.

---

> ### Author Rebuttal · Authors · 2026-03-26
>
> We sincerely thank the reviewer for the thoughtful and constructive feedback. We address each question below.
>
> ## Q1. Preference oracle at intermediate time steps
>
> We would like to first clarify a potential misunderstanding.
> Our method **does not strictly require** step-level preference signals; they can be helpful **but not essential.**
>
> Similar to standard RLHF or RLVR setups, we assume a **single** preference label provided at the trajectory level. Our key modeling choice is to reinterpret this trajectory-level preference as the sum of step-wise regret signals.
> In practice, **our dataset contains no step-level annotations**, and our method does not rely on such fine-grained supervision.
>
> ## Q2. Relation to bandit setting and DPO
>
> As clarified above, we still assumes one preference query per trajectory pair, consistent with practical RLHF pipelines. However, this does not reduce the problem to a standard bandit setting.
>
> The key distinction is that DPO can be understood from a bandit perspective, whereas RePO is formulated from a step-level MDP perspective. This is because RePO explicitly incorporates the behavior policy that generated each trajectory and interprets the future contribution of each step through the lens of regret.
>
> More concretely, RePO does not reduce to DPO:
> - In DPO, the preference signal is applied uniformly across all steps.
> - In contrast, RePO induces **non-uniform** updates by leveraging the **behavior policy** to infer how each step contributes to future regret.
>
> Empirically, Section 7.3 demonstrates a clear performance gap between DPO and RePO. Furthermore, RePO inherently encodes the inductive bias described in Lemma 6.1, leading to improved sample efficiency.
>
> ## Q3. Definition and positioning of regret
>
> Regret is typically defined as the **suboptimality gap** between the value of the current policy and the optimal value, which aligns with our formulation. The key difference from Lattimore \& Szepesvari is that those works consider online settings with repeated rollouts, where regret is accumulated over episodes.
>
> In contrast, our setting is offline, where we only have access to realized trajectories without a simulator. Therefore, it is not meaningful to define regret as a sum over episodes of learning. Instead, our formulation can be interpreted as a sum of immediate regret along a trajectory, consistent with the notion of per-step suboptimality.
> In this sense, our definition is still aligned with standard concepts, but adapted to the offline setting.
>
> ## Q4. Practicality of temporal preference signals
>
> As discussed in Q1 and Q2, we emphasize again that **our method does not strictly require step-level annotations**, and we hope this clarifies the reviewer’s main concern.
>
> At the same time, such annotations are not entirely unrealistic in practice, and our framework can naturally accommodate them when available. For example, in the literature on process reward modeling, preference signals can be assigned at intermediate steps (e.g., identifying the first incorrect reasoning step).
>
> Our initial motivation also included scenarios where preferences may arise intermittently at different stages, reflecting the prospective nature of human judgment. However, constructing such datasets at scale introduces practical challenges, including API cost and consistency issues. Therefore, to isolate the comparison between regret minimization and reward maximization, we use a standard outcome-level preference dataset, consistent with prior work.
>
> ## Q5. On the nature of the theoretical results
>
> We respectfully disagree with the characterization that the theoretical results are merely derivations for the RePO loss.
>
> We intentionally placed purely technical derivations (e.g., Lemma A.1) in the appendix, and included in the main paper only results with broader implications:
>
> - Definition 5.2 and Lemma 5.3 provide a necessary and sufficient condition on the reward structure in KL-constrained RL, enabling a closed-form characterization of optimal policies beyond the regret minimization framework.
> The discussion of $\beta(\cdot)$ (main text and Appendix C) offers insights into the role of reward shaping in existing methods.
> - Theorem 5.4 gives a key representation for expressing off-policy learning objectives in DPO-like forms, by characterizing the $Q$-function in terms of $\pi_{\theta}$ and the behavior policy $\mu$.
> - Lemma 5.5 and 6.1 describe key theoretical properties of RePO, including invariance to $\beta(\cdot)$ (removing the need for reward shaping) and an inductive bias toward verifier-accepted outputs, which explains its empirical sample efficiency.
>
> We hope this clarifies that our theoretical contributions go beyond algorithmic derivation and provide structural insights into preference-based learning.
>
> ---
> We appreciate the reviewer’s careful reading and hope our clarifications address the concerns. We would be happy to further elaborate on any remaining points.

---

> > ### Author Rebuttal · Reviewer_jKZb · 2026-04-01
> >
> > Dear authors,
> >
> > Thank you for your rebuttal.
> >
> > Q1. Thank you for clarifying this.
> >
> > Q2. As a follow-up question, can the method be implemented when the behaviour policy $\mu$ is not known? This is rarely known in practice. For example, when the responses in the preference dataset are generated by bigger models. This is the case, for example, in UltraFeedback.
> >
> > Q3. I still think that regret here would sound confusing for online learning people which invented the terminology, as you say, regret is inherently an online concept. Using such terminology is the offline setting is rather uncommon. I would maybe use the term suboptimality or simple regret [1].
> >
> > Q4. Thank you for the clarification.
> >
> > Q5. I remain convinced that these results are not about convergence properties of RePO, therefore the section should be named algorithmic derivations and not theoretical results. The lack of theory is actually the main weakness of the paper in my opinion.
> >
> > Best,
> > Reviewer
> > [1] Zhao et al. "Revisiting Simple Regret: Fast Rates for Returning a Good Arm".

---

> > > ### Author Response · Authors · 2026-04-02
> > >
> > > Thank you for the follow-up and for clarifying your remaining concerns. We address them below.
> > >
> > > ## Q2. Behavior policy availability
> > >
> > > We agree that access to behavior policy information may be limited in practice, and this is a scenario we explicitly considered in our work.
> > >
> > > In fact, we directly address this setting in Section 6 through **RePO_det**, where each trajectory is treated as if it were generated from a deterministic policy. This yields a pseudo-labeled formulation that removes the need for explicit behavior policy information.
> > >
> > > We further study this scenario empirically in Section 7.2. As shown in Tables 1 and 2, RePO_det performs comparably to RePO, demonstrating that our framework remains robust even when behavior policy information is unavailable. We therefore believe this concern is already addressed in both the methodological design and experimental validation of the paper.
> > >
> > > ## Q3. Regret definition and notation
> > >
> > > We thank the reviewer for pointing this out and apologize for the notational inconsistency.
> > >
> > > Specifically, while Figure 1 correctly presents the objective as a *sum of regret over the trajectory*, the summation $\sum_{t \leq T}$ was omitted in Appendix D.3 for both RePO and RePO_det. Concretely, the corrected objective corresponds to a cumulative (trajectory-level) aggregation of per-step terms:
> > >
> > > $$ -\log \sigma \Big(  \sum\_{t \leq T} \Big[ \alpha \log \frac{\pi\_{\theta}({o^+\_t} \| {q^+\_{<t}} )}{\pi_{\text{ref}}(o^+\_t| q^+\_{<t})} - \alpha \bar{\mathbb{D}}\_{\text{KL}}(\mu{^+} || \pi\_{\theta}; q^+\_{<t}, o^+\_t) \Big] - \sum\_{t \leq T} \Big[ \alpha \log \frac{\pi_\theta(o^-\_t| q^-\_{<t})}{\pi_{\text{ref}}(o^-\_t | q^-\_{<t})} - \alpha \bar{\mathbb{D}}\_{\text{KL}}(\mu^- || \pi\_{\theta}; q^-\_{<t}, o^-\_{t})
> > > \Big] \Big)
> > > $$
> > >
> > > which makes explicit that our objective operates on **trajectory-level aggregation of step-wise quantities**.
> > >
> > > Our per-step quantity does not appear to coincide with the notion of *simple regret* in the online learning literature. In particular, simple regret is typically defined in bandit settings as the difference in immediate utility, whereas our quantity is defined as the difference in **long-term value** between the optimal policy and the behavior policy at each timestep.
> > >
> > > To avoid confusion, we will revise the terminology as follows:
> > >
> > > - we refer to the per-step quantity as **step-wise regret**, and
> > > - we refer to its trajectory-level sum as **cumulative regret**.
> > >
> > > Concretely, we will update Section 4.2 and add the following clarification as a footnote:
> > >
> > > > In the online learning literature, regret is typically defined with respect to instantaneous performance differences. In our setting, the per-step quantity is defined as the gap between the value of the optimal policy and the action-value under the behavior policy, which reflects the suboptimality of future continuation. For clarity, we refer to this as a *step-wise regret*, and its trajectory-level sum as *cumulative regret*.
> > >
> > >
> > > ## Q5. Nature of the theoretical results
> > >
> > > We appreciate the reviewer’s clarification of this concern. We agree that our work does not provide convergence-rate guarantees in the classical online RL sense. However, we emphasize that theoretical contributions in this area are not limited to, nor necessarily require, such guarantees. Moreover, in the offline preference learning setting we consider, convergence analysis in the classical online RL sense is not directly applicable.
> > > Instead, theoretical contributions in RLHF often focus on structural properties of the problem, including equivalence characterizations, invariance results, and closed-form representations of the optimized objective.
> > >
> > > Our Section 5 falls into this category: it characterizes the equivalence class of rewards and Q-functions under KL-regularization, establishes invariance properties of regret-based learning, and derives a behavior-conditioned closed-form representation that connects directly to DPO-style objectives. These results provide structural insight into the preference learning problem, beyond implementation-level derivations.
> > >
> > > We also note that this type of contribution is consistent with prior work in the literature. For example, DPO[1] and KTO[2] both present “theoretical analysis” sections centered on structural results rather than convergence guarantees.
> > > Following the reviewer’s suggestion, we are willing to refine the section title (e.g., to “Theoretical Characterization”) while preserving the distinction from purely algorithmic derivation. We believe this more clearly reflects the nature of our contributions.
> > >
> > > We thank the reviewer again for the helpful feedback. If there are any remaining concerns, we would be happy to further clarify them and will provide updates in this discussion thread.
> > >
> > > ---
> > > [1] Rafailov, Rafael, et al. "Direct preference optimization: Your language model is secretly a reward model."
> > >
> > > [2] Ethayarajh, Kawin, et al. "Kto: Model alignment as prospect theoretic optimization."

---

### Official Review · Reviewer_pVBJ · 2026-03-11

**Soundness:** 3
**Presentation:** 4
**Significance:** 3
**Originality:** 3
**Overall Recommendation:** 5
**Confidence:** 3

**Summary:**

This paper addresses an important issue in RLHF: human judgments are often prospective and counterfactual, yet this characterization has largely been ignored in existing preference-based learning methods. To address this gap, the paper reformulates preference learning in RLHF from a regret-minimization perspective and proposes the RePO framework. Extensive experiments across multiple benchmarks demonstrate the effectiveness of the proposed method.

**Compliance With Llm Reviewing Policy:**

Affirmed.

**Final Justification:**

I maintain my positive score.

**Key Questions For Authors:**

Overall, I find this to be a solid paper and currently lean toward acceptance.

**Limitations:**

Yes

**Strengths And Weaknesses:**

**Strength**

1. The paper is well motivated. It clearly identifies two key limitations of the standard reward-maximization framework: its misalignment with prospective judgment and with counterfactual thinking, and the motivating examples are easy to understand and convincing. From this perspective, the shift from reward maximization to regret minimization feels natural and well justified.

2. The paper is technically solid and provides theoretical results supporting the proposed RePO framework.

3. Extensive experiments across a variety of benchmarks show that RePO consistently improves performance and remains effective across settings.


**Minor issues**

The paper derives a closed-form optimal policy update under the regret-based objective. While the derivation appears mathematically sound, the intuition behind this update could be explained more clearly. For example, what does the optimal solution mean intuitively for model behavior? How does it differ, at an intuitive level, from the policy updates used in DPO- or PPO-style methods? Providing more intuition here would make the theoretical results more accessible to a broader audience.

---

> ### Author Rebuttal · Authors · 2026-03-30
>
> We sincerely thank the reviewer for the thoughtful and constructive feedback. We address each question below.
>
> ## Q1. On the intuition behind the optimal update
>
> At a high level, the key difference is that reward-maximization methods such as DPO or PPO treat each timestep as providing an immediate utility signal, and therefore apply preference supervision uniformly across the trajectory.
> In contrast, RePO evaluates each action based on **how suboptimal its future continuation is relative to the optimal policy.** This is captured by the regret formulation:
>
> $$\text{Reg}\^{\mu}\_{\pi^{\*}}(q\_{<t}, o):= V^{\pi^*}(q_{<t}) - Q^\mu(q_{<t}, o) = -\alpha \left( \log \frac{\pi^{\*}(o \mid q\_{<t})}{\pi\_{\text{ref}}(o \mid q\_{<t})} - \mathbb{D}\_{\text{KL}}(\mu \parallel \pi^{\*}; q\_{<t}, o) \right), $$
> which, under KL-regularization, yields a form involving both a likelihood ratio term and a sequential KL-divergence term.
>
> As a result, the RePO update does not uniformly increase the probability of preferred responses. Instead, it performs a **non-uniform, behavior policy-dependent reweighting** across timesteps based on the set of future trajectories reachable from each point and their degree of suboptimality.
>
> Importantly, RePO is not aimed at a fundamentally different optimal policy from that of DPO. Rather, it provides a more precise and behavior policy-aware training signal. This enables improved alignment with the basis of human preference labels and leads to both stronger performance and improved sample efficiency, as shown in Section 7.3.
>
> We appreciate the reviewer’s suggestion to further improve the intuition behind the update rule. Following the reviewer’s suggestion, we will strengthen the intuitive explanation in the main paper, particularly around Figure 1, Figure 3, and Section 4, to better illustrate how the regret-based update differs from DPO-style updates and how it connects to prospective human judgment.
>
> > For additional visualizations and more detailed results, please refer to our response to Reviewer **vD71** (**Q3**) and [ [this link](https://github.com/Anonymous-account-for-openreview/supplementary-material-for-icml2026/blob/main/images/graph_1_7B_mask_aug_30k.png) ].

---

> > ### Author Rebuttal · Reviewer_pVBJ · 2026-04-02
> >
> > I thank the author for the detailed rebuttal, and I keep my positive score.

---

> > > ### Author Response · Authors · 2026-04-03
> > >
> > > Thank you very much for your positive review.
> > >
> > > We sincerely appreciate your helpful feedback. Your comment on strengthening the empirical results helped us improve the paper, and we will incorporate these additions in the revised version.

---

### Decision · Program_Chairs · 2026-04-30

**Decision:**

Accept (spotlight)

**Comment:**

The paper offers a novel perspective by framing preference learning in RLHF as regret minimization rather than reward maximization, with a closed-form update compatible with DPO. Its central claim—that human preferences are often prospective and counterfactual—is clear and consistently developed. This conceptual shift is a notable strength, and it is supported by both algorithmic design and empirical results.

The experiments are fairly broad and show that RePO and RePO_det are generally competitive with, and often stronger than, standard baselines. This gives the work practical relevance beyond its conceptual contribution.

The main weaknesses are in presentation and positioning. The theoretical part is better understood as characterization and derivation than as providing strong formal guarantees. In addition, the use of the term “regret” in an offline setting may cause confusion, and the assumptions on the behavior policy, as well as the role of RePO_det, would benefit from clearer discussion. While the rebuttal addressed some of these points, a few concerns remain.

Overall, the paper appears positive. It addresses an important problem, proposes an original and coherent viewpoint, and supports it with meaningful empirical evidence. The final version would be strengthened by sharper terminology, clearer theoretical framing, and a more explicit discussion of the practical relationship between RePO and DPO in offline settings.